# REPRESENTATION ALIGNMENT FOR DIFFUSION TRANSFORMERS WITHOUT EXTERNAL COMPONENTS

**Dengyang Jiang**[2,1‡]   **Mengmeng Wang**[3,1*]   **Liuzhuozheng Li**[1]   **Lei Zhang**[2]

**Haoyu Wang**[2]   **Wei Wei**[2]   **Guang Dai**[1]   **Yanning Zhang**[2]   **Jingdong Wang**[4†]

[1]SGIT AI Lab, State Grid Corporation of China  [2]Northwestern Polytechnical University
[3]Zhejiang University of Technology  [4]Baidu Inc.

*Quick overview at:* `https://vvvvvjdy.github.io/sra`

*Code is available at:* `https://github.com/vvvvvjdy/SRA`

## ABSTRACT

Recent studies have demonstrated that learning a meaningful internal representation can accelerate generative training. However, existing approaches necessitate to either introduce an off-the-shelf external representation task or rely on a large-scale, pre-trained external representation encoder to provide representation guidance during the training process. In this study, we posit that the unique discriminative process inherent to diffusion transformers enables them to offer such guidance without requiring external representation components. We propose *Self-Representation Alignment* (**SRA**), a simple yet effective method that obtains representation guidance using the internal representations of learned diffusion transformer. SRA aligns the latent representation of the diffusion transformer in the earlier layer conditioned on higher noise to that in the later layer conditioned on lower noise to progressively enhance the overall representation learning during only the training process. Experimental results indicate that applying SRA to DiTs and SiTs yields consistent performance improvements, and largely outperforms approaches relying on auxiliary representation task. Our approach achieves performance comparable to methods that are dependent on an external pre-trained representation encoder, which demonstrates the feasibility of acceleration with representation alignment in diffusion transformers themselves.

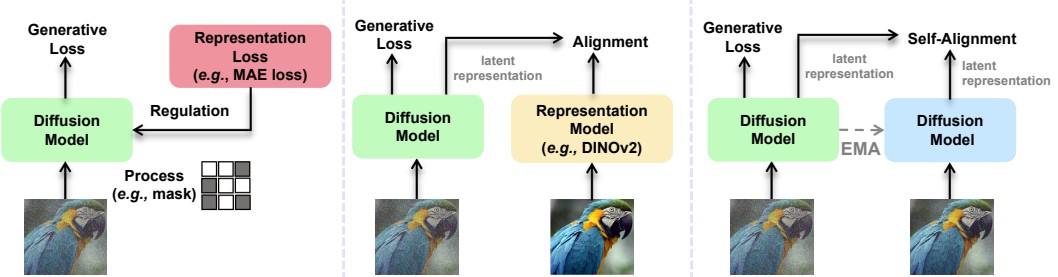

Figure 1: **Left:** Methods like MaskDiT and SD-DiT use an external representation task to guide diffusion transformer. **Middle:** Methods like REPA leverage an external representation foundation model as guidance. **Right (our approach):** We do not use any external representation component but still obtain such guidance through proposed self-representation alignment technique.

## 1 INTRODUCTION

Diffusion transformers (Peebles & Xie, 2023; Ma et al., 2024; Chen et al., 2024a; Team et al., 2025) and vision transformers (Dosovitskiy et al., 2021; Liu et al., 2021; Wang et al., 2021) have held

---

‡This work was completed during the internship at SGIT AI Lab, State Grid Corporation of China.
*Corresponding author. †Project lead.

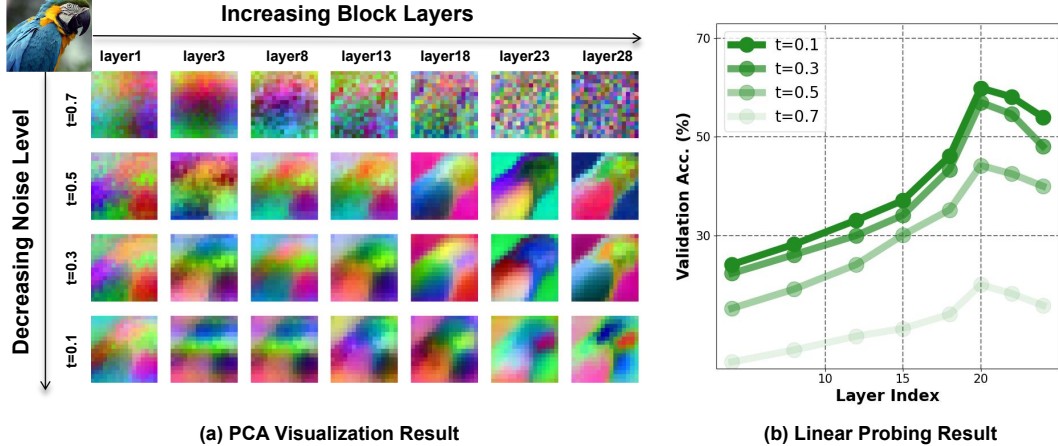

(a) PCA Visualization Result          (b) Linear Probing Result

Figure 2: We empirically investigate the trend of latent representations in the original SiT-XL/2. **Left:** Using PCA Abdi & Williams (2010) to visualize the latent features, we observe that the features lead a process from bad to good when increasing block layers and decreasing noise level. **Right:** A similar trend can also be seen in the linear probing results on ImageNet. Investigation of DiT is provided in Appendix A, which leads to the similar results as SiT.

the dominant positions in visual generation and representation because of their scalability during pre-training (Brooks et al., 2024; Chen et al., 2022; Oquab et al., 2023; Sun et al., 2024) and generalization capacity for downstream tasks (Kirillov et al., 2023; Liu et al., 2024; Lin et al., 2025).

Recently, many works (Zheng et al., 2023; Zhu et al., 2024a; Yu et al., 2025; Krause et al., 2025) have explored leveraging representation components of vision transformers for diffusion transformer's training and have shown that learning a high-quality internal representation can not only speed up the generative training progress but also improve the generation quality. These works either utilize the an external discriminative loss in representation learning (*e.g.*, MAE's (He et al., 2022), IBOT's (Zhou et al., 2022)) shown in Figure 1(left) or leverage a large-scale pre-trained representation foundation model (*e.g.*, DINOv2 (Oquab et al., 2023), CLIP (Radford et al., 2021)) shown in Figure 1(middle) to give representation guidance for the diffusion transformer during the original generative training. However, the former methods require an external representation task, and the latter method relies on a powerful external pretrained encoder, which limits its usage scenarios when there are no good external encoders. Thereout, an intriguing yet underexplored problem has come to light: *Can we obtain such representation guidance without external representation components?*

**Our observations:** Different from the representation model that takes a clean image as input and then outputs semantically-rich feature, diffusion model often takes a noise latent as input and obtains cleaner one step-by-step. In other words, the generative mechanism by which the diffusion model operates can be generally considered as a bad to good process. Inspired by this behavior, we hypothesise that the representations in it also follow such a trend. To testify this, we perform an empirical analysis with recent diffusion transformers (Ma et al., 2024; Peebles & Xie, 2023). As shown in Figure 2(left), we first find out that the latent features in the diffusion transformer are progressively refined, moving from bad to good, as block layers increase and noise level decreases. Next, akin to the results in previous studies (Yu et al., 2025; Xiang et al., 2023), we observe that the diffusion transformer already learns meaningful discriminative representations as shown in Figure 2(right). Meanwhile, although the accuracy drops off after reaching a peak at about layer 20 because the model needs to shift away to focus on generating images with high-frequency details, the quality of the representations basically transfers from bad to good by increasing block layers and decreasing noise level. These indicate that the diffusion transformer gets roughly from a bad-to-good discriminative process when only generative training is performed.

**Our approach:** The observations above motivate us to align the weaker representations in the diffusion transformer to the better ones in the original generative training, thereby not only giving supervision to intermediate layers for fast convergence (Lee et al., 2015), but also enhancing the overall representation learning of the model without involving any external representation component. Driven from this inspiration, we present *Self-Representation Alignment* (SRA). As shown in Figure 1(right), SRA does not need any representation component; in essence, it aligns the output latent representation in earlier layer conditioned on higher noise to that in later layer conditioned on

lower noise to achieve self-representation alignment. Meanwhile, in order to boost the performance, we obtain the target features from another model that shares the same architecture with the trainable model but updates weight by weighted moving average (EMA)[‡]. Furthermore, the student's output latent feature is first passed through the projection layers to conduct a slight nonlinear transformation for better representation extraction and then aligned with the target feature output by the teacher. In a nutshell, our SRA can offer a flexible way to integrate representation guidance without external component needs and architectural modification.

Finally, we conduct comprehensive experiments to evaluate the effect of SRA. After a series of component-wise analyses, we show that SRA brings significant performance improvements to both DiTs (Peebles & Xie, 2023) and SiTs (Ma et al., 2024). Moreover, our ablation study highlights the crucial role of internal representations in the success of SRA, which supports our central hypothesis: *diffusion transformers can achieve representation alignment without external components*.

In summary, our main contributions are as follows:

- We analyze the representations in diffusion transformers and assume that the unique discriminative process makes it possible to achieve representation alignment without external components.
- We introduce SRA, a simple yet effective method that aligns the output latent representation of the diffusion transformers in the earlier layer conditioned on higher noise to that in the later layer conditioned on lower noise to achieve self-representation alignment.
- With our SRA, both DiTs and SiTs achieve sustained training speed acceleration and nontrivial generation performance improvement.

## 2 METHOD

### 2.1 PRELIMINARY: TRAINING OBJECT OF DiT AND SiT

As our method is built upon the denoise-based model (DiT) and flow-based model (SiT), to facilitate a more seamless introduction of SRA in the subsequent part, we now present a brief overview of the training object of these two types of models. We omit the class-condition here for simplicity. We also leave more detailed mathematical descriptions of these types of models in Appendix B.

Denoise-based models learn to transform Gaussian noise into data samples through a step-by-step denoising process. Given a pre-defined forward process that gradually adds noise, these models learn the reverse process to recover the original data.

For data point $\mathbf{x}_0$ from distribution $\mathbf{x}_0 \sim p(\mathbf{x})$, the forward process follows: $q(\mathbf{x}_t|\mathbf{x}_{t-1}) = \mathcal{N}(\mathbf{x}_t; \sqrt{1-\beta_t}\mathbf{x}_0, \beta_t\mathbf{I})$. The model learns to reverse this process using a neural network $\boldsymbol{\epsilon}_{\boldsymbol{\zeta}}(\mathbf{x}_t, t)$ that predicts the noise added at each step. The network is trained using a simple mean squared error objective that measures how well it can predict the noise:

$$\mathcal{L}_{\text{simple}} = \mathbb{E}\mathbf{x}_t, \boldsymbol{\epsilon}, t\Big[||\boldsymbol{\epsilon} - \boldsymbol{\epsilon}_{\boldsymbol{\zeta}}(\mathbf{x}_t, t)||_2^2\Big]. \tag{1}$$

Different from the denoise-based model, the flow-based model aims to learn a velocity field $\mathbf{v}_{\zeta}(\mathbf{x}_t, t)$ that governs a probability flow ordinary differential equation (PF ODE). This PF ODE allows the model to sample data by flowing toward the data distribution. This forward process is defined as:

$$\mathbf{x}_t = \alpha_t\mathbf{x}_0 + \sigma_t\boldsymbol{\epsilon}, \quad \alpha_0 = \sigma_T = 1, \alpha_T = \sigma_0 = 0, \tag{2}$$

where $\mathbf{x}_0 \sim p(\mathbf{x})$ is the data, $\boldsymbol{\epsilon} \sim \mathcal{N}(\mathbf{0}, \mathbf{I})$ is Gaussian noise, and $\alpha_t$ and $\sigma_t$ are monotone decreasing and increasing functions of $t \in [0, T]$, respectively. The PF ODE is given by:

$$\dot{\mathbf{x}}_t = \mathbf{v}_{\boldsymbol{\zeta}}(\mathbf{x}_t, t), \tag{3}$$

where the marginal distribution of this ODE at time $p_t(\mathbf{x})$ matches that of the forward process. To learn the velocity field, the model is trained to minimize the following loss function:

$$\mathcal{L}_{\text{velocity}} = \mathbb{E}_{\mathbf{x}_0, \boldsymbol{\epsilon}, t}\Big[||\mathbf{v}_{\boldsymbol{\zeta}}(\mathbf{x}_t, t) - \dot{\alpha}_t\mathbf{x}_0 - \dot{\sigma}_t\boldsymbol{\epsilon}||^2\Big]. \tag{4}$$

For the sake of simplicity, in the following part, we use generative loss ($\mathcal{L}_{\text{gen}}$) to uniformly represent the two generative training objectives of denoise-based and flow-based methods.

---

[‡]In the following part, we use term 'student' to represent the trainable model and 'teacher' to represent the EMA model in SRA for simplicity.

## 2.2 Self-Representation Alignment

Previous studies have demonstrated that learning good internal representations can both speed up the diffusion transformer's training convergence. In SRA, our insight is aligning the student's latent feature in the earlier layer conditioned on higher noise with that in the later layer conditioned on the lower noise of the teacher to conduct self-representation alignment without requiring any external representation components. As depicted in Figure 3, the goal of our simple training framework is to let the diffusion transformer not only predict noise or velocity but also align with better visual representations from itself. This operation thereby provides a simple way to enhance the representation learning in the diffusion transformer during generative training without the need to design complex representation regulation or introduce external representation foundation models.

Formally, let $f$ be the trainable student model and $f_*$ be the teacher model. Considering input noise latent, timestep, and condition to be $\mathbf{x}_t$, $t$, and $c$. Then, we can obtain the student encoder latent output $\mathbf{y} = f^m(\mathbf{x}_t, t, c) \in \mathbb{R}^{B \times N \times D}$, where $B, N, D > 0$ are the batch-size, number of patches and the embedding dimension for $f$, and $m$ denote the output from the $m^{th}$ layer in $f$. Similarly, the output of the teacher can be expressed as $\mathbf{y}_* = f_*^n(\mathbf{x}_{t-k}, t - k, c) \in \mathbb{R}^{B \times N \times D}$. In our SRA, we set $m \leq n$, $k \geq 0$, and $0 \leq (t - k) < t_{max}{}^*$. Hence, we can conduct self-representation alignment using the teacher's output $\mathbf{y}_*$ and the student's output transform $j_\psi(\mathbf{y}) \in \mathbb{R}^{B \times N \times D}$, where $j_\psi(\mathbf{y})$ is a projection of the student encoder output $\mathbf{y}$ that through a lightweight trainable MLPs head $j_\psi$. Notably, this

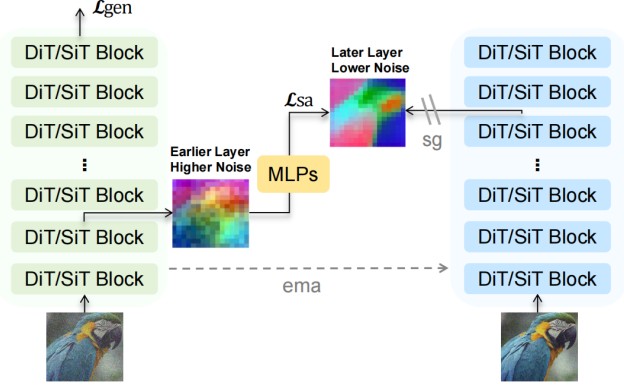

Figure 3: **Overall framework**. SRA aligns the student's latent representation in the earlier layer conditioned on higher noise (green branch) to that of the teacher in the later layer conditioned on lower noise (blue branch) to achieve self-representation alignment. We use a stop-gradient (sg) operator on the teacher to let gradients flow only through the student, and update the teacher's parameters with an exponential moving average (ema) of the student's parameters.

projection head can be discarded optionally after training, which enables SRA to provide guidance without altering any architecture in diffusion transformers.

In particular, SRA attains self-alignment by minimizing the patch-wise distance between the teacher's output ($\mathbf{y}_*$) and the student's output variant ($j_\psi(\mathbf{y})$):

$$\mathcal{L}_{\text{sa}}(\zeta_s, \psi) = \mathbb{E}_{\mathbf{x}_t, t, c}\left[\frac{1}{N} \sum_{i=1}^{N} \text{dist}(\mathbf{y}_*^{[i]}, j_\psi(\mathbf{y}^{[i]}))\right], \tag{5}$$

where $[i]$ is a patch index, $\text{dist}(\cdot, \cdot)$ is a pre-defined distance calculation function, and $\zeta_s$, $\psi$ is the parameters of student diffusion transformer and the projection head.

Finally, we add the abovementioned two objectives for joint learning:

$$\mathcal{L} = \mathcal{L}_{\text{gen}} + \lambda \mathcal{L}_{\text{sa}}, \tag{6}$$

where $\lambda > 0$ is a hyperparameter that controls the trade-off between the generation object and the self-representation alignment object.

## 2.3 Teacher Network

In SRA, we do not need an off-the-shelf teacher to give the guidance. Meanwhile, using the output of the same model as the target to conduct supervision would cause poor results (both found in previous work (Chen & He, 2021; Grill et al., 2020) and our experiment). Thus, we build the

---

*For SiTs, $t_{max} = 1$ and for DiTs, $t_{max} = 1000$. In practice, we truncate $(t - k)$ to 0 if it is less than 0.

teacher from past iterations of the student network using an exponential moving average (EMA) on the student weights. In specific, the updated role of EMA is $\zeta_t = \alpha\zeta_t + (1-\alpha)\zeta_s$, where $\alpha \in [0,1)$ is the momentum coefficient. Note that this is widely used in self-supervised learning (SSL) (Zhou et al., 2022; Caron et al., 2021; He et al., 2020). But we find the default updating role in SSL does not perform well here, instead, $\alpha = 0.9999$ unchanged works surprisingly well in our framework. Experiments and analyses of different $\alpha$ settings are presented in Appendix F. Moreover, we do not use other operations like clustering constraints (Caron et al., 2018; 2020), batch normalizations (Grill et al., 2020; Richemond et al., 2020), and centering (Caron et al., 2021; Zhou et al., 2022) in SSL because we find that the training progress is already stable enough without applying these tricks.

## 3 EXPERIMENT

In this section, we mainly focus on answering the following questions:

- How each design choice and component in SRA influence the performance? (Table 1)
- Does SRA work on different baselines across different model sizes? (Figure 4, and Table 2)
- Can SRA show comparable or superior performance against methods that leverage either the external representation task or the external representation encoder? (Table 3, Table 4, Figure 6)
- Does SRA genuinely enhance the representation capacity of the baseline model, and is the generation capability indeed strongly correlated with the representation guidance? (Figure 7)

### 3.1 EXPERIMENTAL SETUP

**Implementation details.** Unless otherwise specified, the training details strictly follow the setup in DiT (Peebles & Xie, 2023) and SiT (Ma et al., 2024), including AdamW (Loshchilov & Hutter, 2017) with a constant learning rate of 1e-4, no weight decay, batchsize of 256, using the Stable Diffusion VAE (Rombach et al., 2022b) to extract the latent, and etc. For model configurations, we use the B/2, L/2, and XL/2 architectures introduced in the DiT and SiT papers, which process inputs with a patch size of 2. Additional experimental details, hyperparameter settings, linear probing details, and computing resources, are provided in Appendix C.

**Evaluation.** We report Fréchet inception distance (FID (Heusel et al., 2017)), sFID (Nash et al., 2021), inception score (IS (Salimans et al., 2016)), precision (Pre.) and recall (Rec.) (Kynkäänniemi et al., 2019). To ensure a fair comparison with previous methods, we also use the ADM's TensorFlow evaluation suite (Dhariwal & Nichol, 2021)with 50K samples and the same reference statistics. A detailed breakdown of each evaluation metric is included in Appendix D.

**Methods for Comparison.** We compare with recent advanced methods based on diffusion models: (a) *Pixel diffusion*: ADM, VDM++, Simple diffusion, CDM, (b) *Latent diffusion with U-Net*: LDM, and (c) *Latent diffusion with transformers*: DiT, SiT, SD-DiT, MaskDiT, TREAD, REPA, and MAETok. We give detailed descriptions of each method in Appendix E. Note that in the diffusion transformers family, we choose to compare with DiT/SiT and their modifications with representation components involved but do not compare with works aiming at designing advanced architecture like lightningDiT (Yao & Wang, 2025) and DDT (Wang et al., 2025a).

### 3.2 COMPONENT-WISE ANALYSIS

Below, we provide a detailed analysis of the impact of each component. We use SiT-B/2 and train with SRA for 400K iterations for evaluation. Results are shown in Table 1. We also provide some hyperparameter setting principles for more easily transferring SRA to new models in Appendix G.

**Block layers for alignment.** We begin by analyzing the effect of using different blocks of student and teacher for alignment. We observe that using the teacher's last but not least few layers (*e.g.*, 8) to regulate the student's first layers (*e.g.*, 3) leads to optimal performance. We assume that the first few layers need more guidance so they can catch semantically meaningful representation for subsequent generation. Meanwhile, there is a strong correlation between the quality of the representations of the teacher's layers and the performance of the corresponding aligned student (we give the quantitative

Table 1: **Component-wise analysis** on ImageNet 256×256 without classifier-free guidance. ↓ and ↑ indicate whether lower or higher values are better, respectively. $m \rightarrow n$ denotes aligning features from $m^{th}$ layer of the student with that from $n^{th}$ layer of the teacher. $[0, k_{\max})$ denotes time interval $k$ is chosen randomly from 0 to $k_{\max}$. PH. denotes whether to use the projection head.

| Block Layers | Time Interval | $\lambda$ | PH. | FID↓ | IS↑ |
|---|---|---|---|---|---|
| SiT-B/2 Baseline Ma et al. (2024) | | | | 33.02 | 43.71 |
| $6 \rightarrow 10$ | $[0, 0.2)$ | 0.2 | ✓ | 34.85 | 40.28 |
| $4 \rightarrow 8$ | $[0, 0.2)$ | 0.2 | ✓ | 30.00 | 47.78 |
| $4 \rightarrow 10$ | $[0, 0.2)$ | 0.2 | ✓ | 30.65 | 46.69 |
| $4 \rightarrow 12$ | $[0, 0.2)$ | 0.2 | ✓ | 33.19 | 43.30 |
| $2 \rightarrow 6$ | $[0, 0.2)$ | 0.2 | ✓ | 32.14 | 46.36 |
| $2 \rightarrow 8$ | $[0, 0.2)$ | 0.2 | ✓ | 29.31 | 50.13 |
| $3 \rightarrow 8$ | $[0, 0.2)$ | 0.2 | ✓ | **29.10** | **50.20** |
| $3 \rightarrow 3$ | $[0, 0.2)$ | 0.2 | ✓ | 37.08 | 41.54 |
| $3 \rightarrow 8$ | 0.0 | 0.2 | ✓ | 31.07 | 47.32 |
| $3 \rightarrow 8$ | 0.1 | 0.2 | ✓ | 29.55 | 49.01 |
| $3 \rightarrow 8$ | 0.2 | 0.2 | ✓ | 30.70 | 47.72 |
| $3 \rightarrow 8$ | $[0, 0.1)$ | 0.2 | ✓ | 29.38 | 49.32 |
| $3 \rightarrow 8$ | $[0, 0.2)$ | 0.2 | ✓ | **29.10** | **50.20** |
| $3 \rightarrow 8$ | $[0, 0.3)$ | 0.2 | ✓ | 29.15 | 50.01 |
| $3 \rightarrow 8$ | $[0, 0.2)$ | 0.1 | ✓ | 30.65 | 48.31 |
| $3 \rightarrow 8$ | $[0, 0.2)$ | 0.2 | ✓ | **29.10** | **50.20** |
| $3 \rightarrow 8$ | $[0, 0.2)$ | 0.3 | ✓ | 29.28 | 49.72 |
| $3 \rightarrow 8$ | $[0, 0.2)$ | 0.4 | ✓ | 29.75 | 49.30 |
| $3 \rightarrow 8$ | $[0, 0.2)$ | 0.2 | × | 34.23 | 41.07 |
| $3 \rightarrow 8$ | $[0, 0.2)$ | 0.2 | ✓ | **29.10** | **50.20** |

results and analysis in the latter Section 3.4). Based on these results, we set alignment layers as $3 \rightarrow 8$, $6 \rightarrow 16$, and $8 \rightarrow 20$ for B, L, and XL models respectively as default[1]

**Time interval for alignment.** We then study the time interval (meaning the same as $k$ in Section 2.2) used for alignment. Here, we study fixed and dynamic intervals. Notably, we find that using the teacher's features input with a lower noise level than the student's leads to performance enhancement, and an interval value of 0.1 or the mean of 0.1 is optimal. We hypothesize that this is because lower noise levels can offer better representation guidance, but an excessively large time interval can hinder the model's learning process, causing it to focus only on optimizing alignment loss at the expense of neglecting the generative aspects. As dynamic interval shows slightly better performance, we apply time interval as $0 \sim 0.2$ in our feature experiments[2].

**Regularization coefficient for alignment.** We also examine the effect of the coefficient $\lambda$ of self-alignment loss. Note that the results are all better than the baseline, and there is only a little variance for different $\lambda$. We choose 0.2 as default since it achieves overall better performance.

**Effect of projection head for alignment.** We finally examine the effect of the projection head for alignment. Surprisingly, We observe that using this simple head to post-possess the student's output is much better than directly using it to align. We hypothesize this slight operation enables the model to learn more effective hidden representations for subsequent projection head to conduct transformation for final alignment, rather than explicitly aligning the entire latent feature that could potentially disrupt the original generation field that each layer and timestep is responsible for (Zhang et al., 2024; 2023; Frenkel et al., 2024). Hence, we keep using the projection head in future experiments.

### 3.3 SYSTEM-LEVEL COMPARISON

In this section, a system-level comparative study is performed to assess recent diffusion-based approaches against diffusion transformers with SRA.

---

[1]For DiTs, we set alignment layers as $3 \rightarrow 7$, $6 \rightarrow 14$, and $8 \rightarrow 16$ by default because DiT's discriminative behavior across layers is slightly different from SiT's (see Figure 2 and Figure 8).

[2]For DiTs, we set time interval as $\lfloor [0, 200) \rfloor$ by default since DiTs adopt the linear variance schedule with $t$ where $t \in \{0, 1, 2, ..., 999\} \cap \mathbb{Z}$.

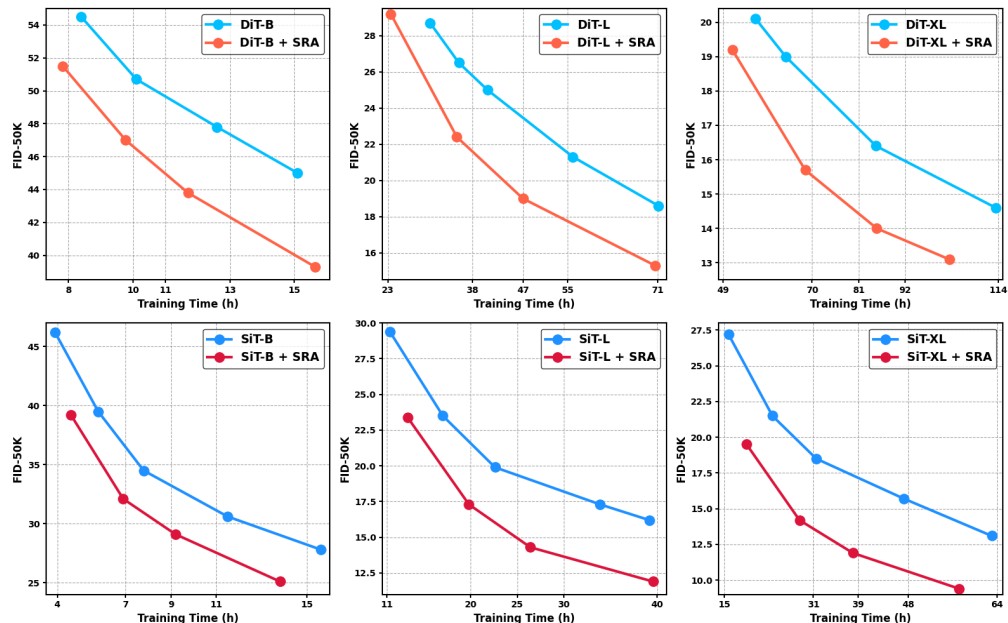

Figure 4: **FID comparisons with vanilla DiTs and SiTs** on ImageNet 256×256 without classifier-free guidance (CFG). The training time (h) is tested on a single machine with 8 A100 GPUs.

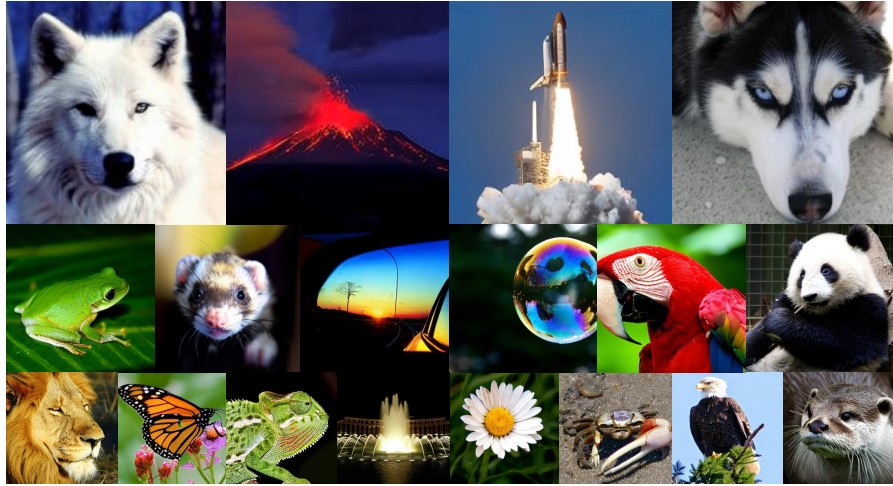

Figure 5: **Selected samples** on ImageNet 256×256 from the SiT-XL + SRA. We use classifier-free guidance with $w = 4.0$. More **uncurated samples** are provided in Appendix L.

**SRA accelerate the convergence of models in different types and sizes.** As our method requires an additional forward pass of the teacher model. For a fair comparison, we report the FID comparison with the baseline under training time. We also give specific training speed and GPU memory usage in Appendix H, as well as FID vs. training time against REPA in Figure 6. As shown in Figure. 4, diffusion transformers trained with SRA demonstrate substantial improvements in performance at the same training time. Moreover, similar to the observation in some SSL works (Oquab et al., 2023; Fan et al., 2025), we notice that the effect of SRA in a larger size model is more significant, which is probably because the larger model tends to provide richer guidance. Meanwhile, the benefits of SRA do not saturate even when the models have already achieved a low FID score. We assume that this is likely due to the teacher's constantly improving capacity, which allows it to provide better and better representation guidance for the student when the training goes on.

Meanwhile, we also conduct text-to-image experiment following REPA to use COCO2014 (Lin et al., 2014) and MMDiT (Esser et al., 2024) to further demonstrate the effect of SRA, the results in Table 2 shows that SRA can naturally extend to text-to-image generation. Our self-representaion

Table 3: **System-level comparison** on ImageNet 256×256 with Classifier-free Guidance (CFG). The **best** and second-best results on each metric are highlighted in bold and underlined.

| Model | Epochs | Tokenizer | FID↓ | sFID↓ | IS↑ | Pre.↑ | Rec.↑ |
|---|---|---|---|---|---|---|---|
| *Pixel diffusion* | | | | | | | |
| ADM-U | 400 | - | 3.94 | 6.14 | 186.7 | 0.82 | 0.52 |
| VDM++ | 560 | - | 2.40 | - | 225.3 | - | - |
| Simple diffusion | 800 | - | 2.77 | - | 211.8 | - | - |
| CDM | 2160 | - | 4.88 | - | 158.7 | - | - |
| *Latent diffusion, U-Net* | | | | | | | |
| LDM-4 | 200 | LDM-VAE | 3.60 | - | 247.7 | **0.87** | 0.48 |
| *Latent diffusion, Transformer* | | | | | | | |
| DiT-XL/2 | 1400 | SD-VAE | 2.27 | 4.60 | 278.2 | 0.83 | 0.57 |
| SiT-XL/2 | 1400 | SD-VAE | 2.06 | **4.50** | 270.3 | 0.82 | 0.59 |
| SD-DiT | 480 | SD-VAE | 3.23 | - | - | - | - |
| MaskDiT | 1600 | SD-VAE | 2.28 | 5.67 | 276.6 | 0.80 | 0.61 |
| DiT + TREAD | 740 | SD-VAE | 1.69 | 4.73 | 292.7 | 0.81 | 0.63 |
| SiT + REPA | 800 | SD-VAE | **1.42** | 4.70 | 305.7 | 0.80 | **0.65** |
| SiT + MAETok | 800 | MAE-Tok | 1.67 | - | 311.2 | - | - |
| **SiT + SRA (ours)** | **400** | SD-VAE | 1.85 | **4.50** | 297.2 | 0.82 | 0.61 |
| **SiT + SRA (ours)** | **800** | SD-VAE | 1.58 | 4.65 | **311.4** | 0.80 | 0.63 |

Table 4: **System-level comparison** on ImageNet 512×512 with Classifier-free Guidance (CFG).

| Model | Epochs | FID↓ | sFID↓ | IS↑ | Pre.↑ | Rec.↑ |
|---|---|---|---|---|---|---|
| *Pixel diffusion* | | | | | | |
| VDM++ | - | 2.65 | - | 278.1 | - | - |
| Simple diffusion | 800 | 4.28 | - | 171.0 | - | - |
| *Latent diffusion, Transformer* | | | | | | |
| DiT-XL/2 | 600 | 3.04 | 5.02 | 240.8 | 0.84 | 0.54 |
| SiT-XL/2 | 600 | 2.62 | 4.18 | 252.2 | **0.84** | 0.57 |
| MaskDiT | 600 | 2.50 | 5.10 | 256.3 | 0.84 | 0.56 |
| SiT + REPA | 200 | **2.08** | 4.19 | 274.6 | 0.83 | 0.58 |
| **SiT + SRA (ours)** | **200** | 2.17 | **4.15** | **279.3** | 0.83 | **0.59** |

alignment strategy works, as evidenced by our experiments: applying SRA to MMDiT without any hyperparameter tuning improves FID ( from 5.66 to 4.75) and PickScore (from 20.65 to 21.14) over the baseline, and is comparable to REPA.

**SRA demonstrates superior or comparable performance compared to other methods.** We also provide a quantitative comparison between SiT-XL with SRA and other recent methods shown in Table 3 and Table 4. In ImageNet 256×256, Our method already outperforms the original SiT-XL model with 1000 fewer epochs, and it is further improved with longer training. At 800 epochs, SRA achieves FID of 1.58 and IS of 311.4. It is worth noting that this result is far superior to methods (*e.g.*, MaskDiT) that rely on external representation task and is comparable with methods (*e.g.*, REPA) that depend heavily on an external pre-trained representation model. In higher resolution settings, the performance of SRA surpasses its baseline using 3× fewer training iterations. Training with the same iterations, SRA outperforms REPA in terms of three metrics (sFID, IS, and Rec) and on par with REPA in other metrics.

Table 2: FID and Pickscore (Kirstain et al., 2023) comparisons with vanilla MMDiT and REPA on COCO2014.

| Method | FID↓ | PickScore↑ |
|---|---|---|
| *ODE, NFE=50, Trained for 150K iter* | | |
| MMDiT | 5.86 | 20.05 |
| MMDiT + REPA | 4.60 | 20.88 |
| MMDiT + SRA | 4.85 | 21.14 |

**SRA shows a more sustained boost than REPA.** As shown in Figure 6, although REPA converges quickly at the beginning of training due to its large-scale, pre-trained representation model for guidance, the performance saturates after about 200 epochs. On the contrary, the progressively higher-quality guidance teacher have throughout training in SRA makes our method to provide a more sustained boost without the need for any external models. This observation of saturation effects and potential detrimental impacts of REPA in later stages has also been corroborated by several

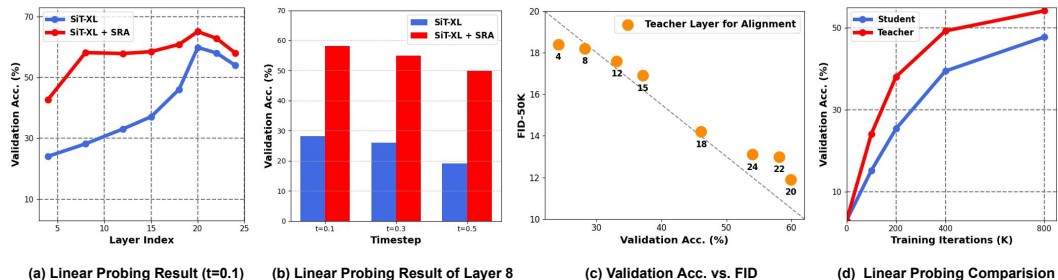

(a) Linear Probing Result (t=0.1)  (b) Linear Probing Result of Layer 8  (c) Validation Acc. vs. FID  (d) Linear Probing Comparision

Figure 7: We empirically investigate the effect of representations in SRA. **(a) and (b):** Linear probing result of vanilla SiT-XL trained for 1400 epochs and SiT-XL + SRA trained for 800 epochs. **(c):** Linear probing vs. FID plot of SiT-XL + SRA with different teacher's output layers for alignment (similar plot of DiT + SRA is provided in Appendix J). **(d):** Linear probing results of layers of the teacher and the student used for alignment during training.

contemporary studies (Wang et al., 2025b; Zhang et al., 2025). We believe this limitation of REPA can be offset by the sustained enhancement facilitated by SRA.

## 3.4 ABLATION STUDY

Since SRA introduces representation guidance in an implicit way, we thus aim at testing whether representation truly matters in SRA. We give the answer with the following experiments.

**Enhanced representation capacity with SRA.** We first compare the representation capacity of vanilla SiT and SiT trained with SRA. As shown in Figure 7((a)) and Figure 7((b)),

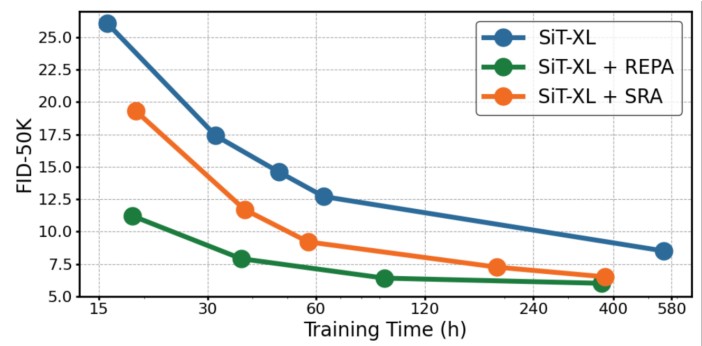

Figure 6: **Training time (h) vs. FID plot** without CFG.

SRA consistently improve the quality of latent representation in the diffusion transformer, as indicated by better linear probing results across different blocks and timesteps.

**Tight coupling between generation quality and representation guidance in SRA.** We then investigate the correlation between generation performance and the representation guidance (detailed experimental setup can be found in Appendix I) in SRA. Figure 7((c)) reveals a strong correlation between linear probing accuracy and FID scores as the teacher network layers for alignment are varied. This finding underscores that the model's generative capabilities are indeed closely tied to the effectiveness of the self-representation guidance mechanism.

**Consistent representation improvement during training in SRA.** We finally verify whether the representational capacity of the student and teacher models is continuously improving during the training process and whether the teacher can always give better guidance for the student. As shown in Figure 7((d)), the teacher's representation quality consistently improves throughout training (from 38.1 at 200K iterations to 54.2 at 800K iterations) and consistently outperforms the student model at each stage. This empirical evidence directly supports our claim that the teacher's improving capacity provides better representation guidance as training proceeds.

## 4 RELATED WORK

Here, we highlight key related studies and defer a discussion of other relevant studies to Appendix K.

**Representations guidance for diffusion transformer's training.** Many recent works have attempted to introduce representation guidance in diffusion transformer training. MaskDiT (Zheng et al., 2023) and SD-DiT (Zhu et al., 2024a) add MAE's (He et al., 2022) and IBOT's (Zhou et al., 2022) learning task into the original DiT's training progress. TREAD (Krause et al., 2025) de-

sign a token routing strategy with MAE loss to speed up the diffusion model's training. REPA (Yu et al., 2025) utilizes a large-scale data pre-trained representation model for regulating the diffusion model's latent feature. VA-VAE (Yao & Wang, 2025) and MAETok (Chen et al., 2025) align the latent distribution of the tokenizer with the external representation foundation model and show this alignment is beneficial to resulting diffusion models. Different from this work, we aim to look for representation guidance in the diffusion model itself and its own training paradigm.

## 5 LIMITATION AND FUTURE WORK

Due to limitations in computation resources, we are unable to conduct large-scale text-to-video (t2v) pretraining with SRA. However, we maintain that applying SRA to video domain is conceptually sound for the following reasons. First, even within the image domain (already exist strong encoder like DINOv2), our method achieves performance comparable to that of REPA, this supports our confidence in the effectiveness of approach in text-to-video generation, where currently there is no well-pretrained strong encoder for open-domain videos (there is some video pretrained model, e.g., VideoMAE (Wang et al., 2023), which is still not strong enough for open-domain video encoder, compared to DINOv2 for image encoder[3].). Second, in the video domain, numerous studies (Wiedemer et al., 2025; Zhu et al., 2024b) have shown that large-scale pretrained t2v models already capture rich and transferable representations suitable for various understanding tasks, further reinforcing the potential of SRA in video-related applications.

Moreover, as mentioned in FLUX.2 (Labs, 2025), when scaling to larger, more diverse data distributions beyond ImageNet and COCO, the approach that does not align with an external representation encoders like DINOv2 shows superior generation quality compared to REPA. This suggests that SRA, which does not rely on external encoders, may have better scalability and generalization potential when applied to more complex, large-scale generation tasks, as the capacity of external encoders may be fundamentally limited by their pre-training data distribution and thus fail to provide meaningful guidance when the target domain exceeds their representational coverage.

## 6 CONCLUSION

In this study, we show that diffusion transformers can provide representation guidance by themselves to boost generation performance with our proposed SRA, which aligns their latent representation in the earlier layer conditioned on higher noise to that in the later layer conditioned on lower noise to progressively enhance the representation learning without external components. Considering the simplicity and effectiveness of SRA, we believe it will facilitate more future research to extend SRA in other scenarios.

## 7 ACKNOWLEDGMENT

This work was supported by Zhejiang Province Natural Science Foundation of China under Grant No. LQN25F030008 and the National Natural Science Foundation of China under Grant No. 62403429.

This manuscript referred to some color schemes and structural options from REPA (Yu et al., 2025) manuscript. We also want to thank Sihyun Yu, Sizhe Dang, and Zanyi Wang for the helpful discussions and suggestions during the progress of SRA project.

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

## ETHICS STATEMENT

This manuscript does not involve any topic and experiment related to ethics.

## REPRODUCIBILITY STATEMENT

This manuscript provides a comprehensive description of the implementation details and hyperparameter settings. The associated code is made publicly available for reproducing our results.

## USE OF LLMS

In this paper, large language models (LLMs) act as collaborative editors to polish the paper, elevating both the precision and readability of our manuscript.

## APPENDIX

## A   INVESTIGATION OF REPRESENTATIONS IN DiT

We also perform a similar analysis with DiT like those have done in Figure 2(left) (PCA visualization) and Figure 2(right) (linear probing), the results are showed in Figure 8. In short, we also observe that the representations in DiT basically lead a process from coarse to fine when increasing block layers and decreasing noise level as SiT's.

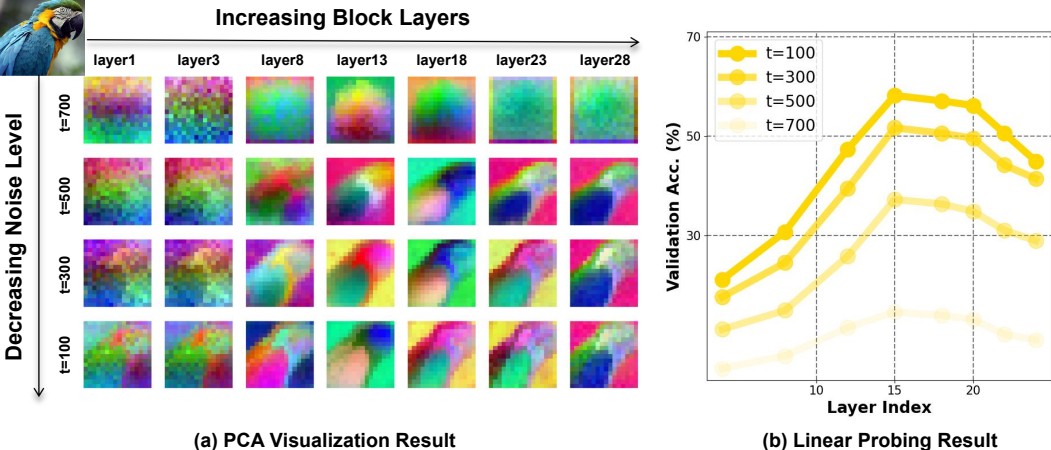

(a) PCA Visualization Result          (b) Linear Probing Result

Figure 8: We also empirically investigate the representations in diffusion transformers across different blocks and timesteps with the original DiT-XL/2 checkpoint trained for 7M iterations. Similar to SiT, the latent representations in DiT basically follow the bad to good process, as block layers increase and noise level decreased.

## B   DESCRIPTIONS FOR TWO TYPES OF BASELINE MODELS

In this paper, we use DiT and SiT as our baseline models. We now provide an overview of two types of generative models that are variants of denoising autoencoders and are used to learn the target distribution. Specifically, we discuss Denoising Diffusion Probabilistic Models (DDPMs) like DiT in Section B.1 and Stochastic interpolant models like SiT in Section B.2.

### B.1   DENOISING DIFFUSION PROBABILISTIC MODELS (DiT)

Denoise-based models (Ho et al., 2020; Nichol & Dhariwal, 2021) aim to model the target distribution ($\mathbf{x} \sim p(\mathbf{x})$) by learning a gradual denoising process that transforms a Gaussian distribution (

$\mathcal{N}(\mathbf{0}, \mathbf{I})$ ) into $p(\mathbf{x})$. Formally, diffusion models learn a reverse process ( $p(\mathbf{x}_{t-1}|\mathbf{x}_t)$ ) corresponding to a pre-defined forward process ( $q(\mathbf{x}_t|\mathbf{x}_0)$ ), which incrementally adds Gaussian noise to the data starting from $p(\mathbf{x})$ over a sequence of time steps ( $t \in 1, \ldots, T$ ), with $T > 0$ fixed.

For a given $\mathbf{x}_0 \sim p(\mathbf{x}_0)$, the forward process ( $q(\mathbf{x}_t|\mathbf{x}_{t-1})$ ) is defined as: $q(\mathbf{x}_t|\mathbf{x}_{t-1}) = \mathcal{N}(\mathbf{x}_t; \sqrt{1 - \beta_t}\mathbf{x}_0, \beta_t^2\mathbf{I})$, where $\beta_t \in (0, 1)$ are pre-defined small hyperparameters. DDPM (Ho et al., 2020) formalizes the reverse process $p(\mathbf{x}_{t-1}|\mathbf{x}_t)$ as:

$$p(\mathbf{x}_{t-1}|\mathbf{x}_t) = \mathcal{N}\Big(\mathbf{x}_{t-1}; \frac{1}{\sqrt{\alpha_t}}\big(\mathbf{x}_t - \frac{\sigma_t^2}{\sqrt{1 - \bar{\alpha}_t}}\boldsymbol{\epsilon}_\zeta(\mathbf{x}_t, t)\big), \boldsymbol{\Sigma}_\zeta(\mathbf{x}_t, t)\Big), \qquad (7)$$

where $\alpha_t = 1 - \beta_t$, $\bar{\alpha}t = \prod i = 1^t \alpha_i$, and $\boldsymbol{\epsilon}_\zeta(\boldsymbol{x_t}, t)$ is parameterized by a neural network.

The model is trained using a simple denoising autoencoder objective:

$$\mathcal{L}_{\text{simple}} = \mathbb{E}_{\mathbf{x}_t, \boldsymbol{\epsilon}, t}\Big[||\boldsymbol{\epsilon} - \boldsymbol{\epsilon}_\zeta(\mathbf{x}_t, t)||_2^2\Big]. \qquad (8)$$

For the covariance term ( $\boldsymbol{\Sigma}_\zeta(\mathbf{x}_t, t)$ ), DDPM(Ho et al., 2020) demonstrated that setting it as ( $\sigma_t^2\mathbf{I}$ ) with ( $\beta_t = \sigma_t^2$ ) is sufficient. Subsequently, Improved-DDPM(Nichol & Dhariwal, 2021) showed that performance can be enhanced by jointly learning ( $\boldsymbol{\Sigma}_\zeta(\mathbf{x}_t, t)$ ) along with ( $\boldsymbol{\epsilon}_\zeta(\mathbf{x}_t, t)$ ) in a dimension-wise manner using the following objective:

$$\mathcal{L}_{\text{vlb}} = \exp(v \log \beta_t + (1 - v) \log \tilde{\beta}_t), \qquad (9)$$

where $v$ is a component per model output dimension, and $\tilde{\beta}t = \frac{1 - \bar{\alpha}t - 1}{1 - \bar{\alpha}_t}\beta_t$.

With a sufficiently large $T$ and an appropriate scheduling of $\beta_t$, the distribution $p(\mathbf{x}_T)$ approaches an isotropic Gaussian distribution. Thus, sampling is achieved by starting from random Gaussian noise and iteratively applying the reverse process ( $p(\mathbf{x}_{t-1}|\mathbf{x}_t)$ ) to recover a data sample $\mathbf{x}_0$ (Ho et al., 2020).

## B.2 STOCHASTIC INTERPOLANTS MODELS (SIT)

Unlike DDPMs, flow-based models (Liu et al., 2023; Lipman et al., 2022) describe a continuous time-dependent process involving data ( $\mathbf{x}_* \sim p(\mathbf{x})$ ) and Gaussian noise ( $\epsilon \sim \mathcal{N}(\mathbf{0}, \mathbf{I})$ ) over $t \in [0, 1]$:

$$\mathbf{x}_t = \alpha_t\mathbf{x}_0 + \sigma_t\boldsymbol{\epsilon}, \quad \alpha_0 = \sigma_1 = 1, \; \alpha_1 = \sigma_0 = 0, \qquad (10)$$

with $\alpha_t$ and $\sigma_t$ being decreasing and increasing functions of $t$, respectively. The process is governed by a probability flow ordinary differential equation (PF ODE):

$$\dot{\mathbf{x}}_t = \mathbf{v}(\mathbf{x}_t, t), \qquad (11)$$

where the distribution of the ODE at time $t$ matches the marginal distribution $p_t(\mathbf{x})$.

and can be approximated by a model $\mathbf{v}_\zeta(\mathbf{x}_t, t)$ trained to minimize the objective:

$$\mathcal{L}_{\text{velocity}} = \mathbb{E}_{\mathbf{x}_0, \boldsymbol{\epsilon}, t}\Big[||\mathbf{v}_\zeta(\mathbf{x}_t, t) - \dot{\alpha}_t\mathbf{x}_0 - \dot{\sigma}_t\boldsymbol{\epsilon}||^2\Big]. \qquad (12)$$

This also corresponds to a reverse stochastic differential equation (SDE) given by:

$$d\mathbf{x}_t = \mathbf{v}(\mathbf{x}_t, t)dt - \frac{1}{2}w_t\mathbf{s}(\mathbf{x}_t, t)dt + \sqrt{w_t}d\bar{\mathbf{w}}_t, \qquad (13)$$

As shown in (Albergo & Vanden-Eijnden, 2023), any functions $\alpha_t$ and $\sigma_t$ that satisfy the following three conditions:

$$1. \; \alpha_t^2 + \sigma_t^2 > 0, \; \forall t \in [0, 1]$$
$$2. \; \alpha_t \text{ and } \sigma_t \text{ are differentiable}, \; \forall t \in [0, 1]$$
$$3. \; \alpha_1 = \sigma_0 = 0, \; \alpha_0 = \sigma_1 = 1,$$

lead to an unbiased interpolation process between $\mathbf{x}_0$ and $\epsilon$. Example choices include linear interpolants ($\alpha_t = 1 - t$, $\sigma_t = t$) or variance-preserving (VP) interpolants ($\alpha_t = \cos(\frac{\pi}{2}t)$, $\sigma_t = \sin(\frac{\pi}{2}t)$) (Ma et al., 2024).

An advantage of stochastic interpolants is that the diffusion coefficient ( $w_t$ ) can be independently selected during sampling with the reverse SDE, even after training. This capability simplifies the design space compared to score-based diffusion models (Karras et al., 2024).

Table 5: **Default hyperparameter setup.** Unless other otherwise specified, we use these sets of hyperparameters for different models. In our component-wise analysis experiment, settings are also kept the same except those we point out in Table 1.

| | SiT-B | SiT-L | SiT-XL | DiT-B | DiT-L | DiT-XL |
|---|---|---|---|---|---|---|
| **Architecture** | | | | | | |
| Input dim. | $32\times32\times4$ | $32\times32\times4$ | $32\times32\times4$ | $32\times32\times4$ | $32\times32\times4$ | $32\times32\times4$ |
| Patch size | 2 | 2 | 2 | 2 | 2 | 2 |
| Num. layers | 12 | 24 | 28 | 12 | 24 | 28 |
| Hidden dim. | 768 | 1024 | 1152 | 768 | 1024 | 1152 |
| Num. heads | 12 | 16 | 16 | 12 | 16 | 16 |
| **SRA** | | | | | | |
| Alignment blocks | $3 \to 8$ | $6 \to 16$ | $8 \to 20$ | $3 \to 7$ | $6 \to 14$ | $8 \to 16$ |
| Alignment time interval | $[0, 0.2)$ | $[0, 0.2)$ | $[0, 0.2)$ | $\lfloor[0, 200)\rfloor$ | $\lfloor[0, 200)\rfloor$ | $\lfloor[0, 200)\rfloor$ |
| Objective | smooth-$\ell_1$ | smooth-$\ell_1$ | smooth-$\ell_1$ | smooth-$\ell_1$ | smooth-$\ell_1$ | smooth-$\ell_1$ |
| EMA decay | 0.999 | 0.999 | 0.999 | 0.999 | 0.999 | 0.999 |
| Using projection head | ✓ | ✓ | ✓ | ✓ | ✓ | ✓ |
| **Optimization** | | | | | | |
| Batch size | 256 | 256 | 256 | 256 | 256 | 256 |
| Optimizer | AdamW | AdamW | AdamW | AdamW | AdamW | AdamW |
| lr | 0.0001 | 0.0001 | 0.0001 | 0.0001 | 0.0001 | 0.0001 |
| $(\beta_1, \beta_2)$ | (0.9, 0.999) | (0.9, 0.999) | (0.9, 0.999) | (0.9, 0.999) | (0.9, 0.999) | (0.9, 0.999) |
| **Interpolants or Denoising** | | | | | | |
| $\alpha_t$ | $1 - t$ | $1 - t$ | $1 - t$ | - | - | - |
| $\sigma_t$ | $t$ | $t$ | $t$ | - | - | - |
| $w_t$ | $\sigma_t$ | $\sigma_t$ | $\sigma_t$ | - | - | - |
| T | - | - | - | 1000 | 1000 | 1000 |
| Training objective | v-prediction | v-prediction | v-prediction | noise-prediction | noise-prediction | noise-prediction |
| Sampler | Euler-Maruyama | Euler-Maruyama | Euler-Maruyama | DDPM | DDPM | DDPM |
| Sampling steps | 250 | 250 | 250 | 250 | 250 | 250 |
| Guidance Scale | - | - | 1.8 (if used) | - | - | - |

## C  HYPERPARAMETER AND MORE IMPLEMENTATION DETAILS

**More implementation details.** We implement our models based on the original DiT and SiT implementation. To speed up training and save GPU memory, we use mixed-precision (fp16) with a gradient clipping and FusedAttention (Dao, 2023) operation for attention computation. We also pre-compute compressed latent vectors from raw pixels via stable diffusion VAE (Rombach et al., 2022b) and use these latent vectors. We do not apply any data augmentation, but we find this does not lead to a big difference, as similarly observed in MaskDiT (Zheng et al., 2023) and REPA (Yu et al., 2025). We also use `stabilityai/sd-vae-ft-ema` for encoding images to latent vectors and decoding latent vectors to images. For the projection head used for nonlinear transformation, we use two-layer MLP with SiLU activations (Elfwing et al., 2018). We use smooth-$\ell_1$ objective for alignment because we find it perform slightly better than $\ell_2$ and its gradient transitions are smoother than $\ell_1$. When using SiT-XL to generate images with classifier-free guidance (Ho & Salimans, 2022), the guidance interval introduced in the study (Kynkäänniemi et al., 2025) with the same setting used in REPA (Yu et al., 2025) is applied, which has been demonstrated to yield a slight performance improvement.

**Sampler.** For DiT, we use the DDPM sampler and set the number of function evaluations (NFE) as 250 by default. For SiT, we use the SDE Euler-Maruyama sampler (for SDE with $w_t = \sigma_t$) and set the NEF as 250 by default. The settings also align with those used in DiT and SiT papers.

**Dataset.** We use ImageNet (Deng et al., 2009), where each image is preprocessed to the resolution of 256×256 or 512×512 (denoted as ImageNet 256×256 or ImageNet 512×512), and follow ADM (Dhariwal & Nichol, 2021) for other data preprocessing protocols.

**Linear probing.** We follow the setup used in REPA (Yu et al., 2025) and I-DAE (Chen et al., 2024b), and use the code-base of ConvNeXt (Liu et al., 2022) to conduct the experiment. Specifically, we use AdaptiveAvgPooling and a batch normalization layer to process the output of the latent feature by the model, then train a linear layer for 80 epochs. The batch size is set to 4096 with a cosine decay learning rate scheduler, where the initial learning rate is set to 0.001.

**Computing resources.** We use 8 NVIDIA A100 80GB GPUs or 8 NVIDIA L40S 48GB GPUs for training largest model (XL); and use 4 NVIDIA A100 80GB GPUs or 4 NVIDIA L40S 48GB GPUs for training smaller model (L, B), our training speed is about 2.12 step/s with a global batch size of 256. When sampling, we use either NVIDIA A100 80GB GPUs or NVIDIA L40S 48GB GPUs or NVIDIA RTX 4090 24GB GPUs to obtain the samples for evaluation.

## D  EVALUATION METRIC

In this section, we define the key metrics used to assess the performance of our model. Each metric is summarized below:

- **Fréchet Inception Distance (FID) (Heusel et al., 2017)**
  Purpose: Measures the similarity between the feature distributions of real and generated images.
  Methodology: Uses the Inception-v3 network (Szegedy et al., 2016) to extract features. Assumes both feature distributions are multivariate Gaussian and computes the Fréchet distance between them.
  Interpretation: Lower FID scores indicate better similarity (and thus higher-quality generated images).

- **Spatial Fréchet Inception Distance (sFID) (Nash et al., 2021)**
  Purpose: Extends FID by incorporating spatial information to better capture the structural fidelity of generated images.
  Methodology: Computes FID using intermediate spatial features (rather than global features) from the Inception-v3 network.

- **Inception Score (IS) (Salimans et al., 2016)**
  Purpose: Evaluates the quality and diversity of generated images.
  Methodology: Uses the Inception-v3 network to compute class probabilities (logits) for generated images. Measures the KL-divergence between the marginal class distribution of generated images and the conditional class distribution of a single image (after softmax normalization).
  Interpretation: Higher IS values indicate both high image quality (confident predictions) and diversity (uniform marginal distribution).

- **Precision and Recall for Distributions (Precision/Recall) (Kynkäänniemi et al., 2019)**
  Purpose: Evaluates the trade-off between sample quality (precision) and distribution coverage (recall).
  Methodology: Precision: Fraction of generated images deemed realistic by a classifier (relative to real images). Recall: Fraction of the real image manifold covered by generated samples.

## E  METHODS FOR COMPARISON

Next, we explain the key ideas behind the methods used for evaluation and comparison.

- **ADM** (Dhariwal & Nichol, 2021) enhances U-Net-based architectures for diffusion models and introduces classifier-guided sampling, a technique used to balance the quality-diversity tradeoff and improve overall performance.

- **VDM++** (Kingma & Gao, 2023) improves diffusion model training efficiency through an adaptive noise scheduling mechanism that dynamically adjusts noise levels during optimization.

- **Simple Diffusion** (Hoogeboom et al., 2023) targets high-resolution synthesis by simplifying both network architectures and noise schedules through systematic exploration of lightweight design choices.

- **CDM** (Ho et al., 2022) achieves high-fidelity generation via a cascaded pipeline: training low-resolution diffusion models first, then progressively refining details through super-resolution diffusion stages.

- **LDM** (Rombach et al., 2022a) accelerates training while maintaining generation quality by operating in compressed latent spaces, modeling image distributions at reduced dimensionality compared to pixel space.

- **DiT** (Peebles & Xie, 2023) employs a pure transformer backbone for diffusion models, incorporating AdaIN-zero modules as an important component modification against vanilla vision transformer (Dosovitskiy et al., 2021) of its architecture.

- **SiT** (Ma et al., 2024) provides an extensive analysis of how the training efficiency of DiTs can be improved by transitioning from discrete diffusion to continuous flow-based modeling, which can be seen as a flow-based version of DiT.

- **SD-DiT** (Zhu et al., 2024a) leverages IBOT's (Zhou et al., 2022) training paradigm that combines DINO loss (Caron et al., 2021) and BEIT loss (Bao et al., 2022) for efficiently training diffusion transformers.

- **MaskDiT** (Zheng et al., 2023) proposes an asymmetric encoder-decoder scheme for efficient training of diffusion transformers, where they train the model with an auxiliary mask reconstruction task similar to MAE (He et al., 2022).

- **TREAD** (Krause et al., 2025) introduces a dynamic token routing strategy combined with the mask reconstruction task similar to MAE (He et al., 2022) and MaskDiT (Zheng et al., 2023) to accelerate the training of diffusion models.

- **REPA** (Yu et al., 2025) achieves significant improvements in both training efficiency and generation quality by aligning the latent feature of the diffusion model with that of a large-scale data pre-trained representation model (*e.g.*, DINOv2 (Oquab et al., 2023)).

- **MAETok** changes the SD-VAE to MAE-Tok, which is trained with auxiliary mask reconstruction loss and aligns loss with three representation targets (HOG's (Dalal & Triggs, 2005), DINOv2's (Oquab et al., 2023), and CLIP's (Radford et al., 2021)) and obtains diffusion transformer with better generation performance.

## F    TEACHER NETWORK UPDATING ROLE

Table 6: Different EMA update rule attempts, other settings are consistent with the default setting of Table 1 in the main paper.

| $\alpha$ of EMA | FID↓ | IS↑ |
|---|---|---|
| 0.0 | 35.71 | 42.18 |
| $0.996 \rightarrow 1.0$ | 33.17 | 44.96 |
| 0.9999 | **29.10** | **50.20** |

In other generative learning studies, the EMA model is often used solely for evaluation. However, as we need it to provide guidance during training, we study different updating methods. Here, we investigate three different strategies to build the teacher. First, we find that using teacher copied from a student ($\alpha = 0$) would impair the performance, which testifies to our argument in Section 1 . Next, we consider using the strategy widely used in self-supervised learning works (Caron et al., 2021; Zhou et al., 2022) that updates the momentum coefficient $\alpha$ from 0.996 to 1 during training. However, in our framework, this does not work well. Finally, we use the $\alpha$ of 0.9999 unchanged, which is commonly used in other generative learning works (Dhariwal & Nichol, 2021; Peebles & Xie, 2023) and find it is also suitable for our framework. In our analysis, first, student copy ($\alpha = 0.0$) performs the worst : this is consistent with the observation in self-supervised learning: student copy leads to unstable training and cannot converge; there is a diffusion loss in our approach, so the training still converges. Second, using $\alpha = 0.9999$ is better than $0.996 \rightarrow 1.0$: In EMA, a larger $\alpha$ indicates a smoother update. We analyze that the reason for keeping such a very large value of $\alpha$ works well in SRA is that: in our method, the enhanced representation learning is ultimately for the service of generation. Thus, when the update of teacher model is relatively intense, the alignment would be less stable, the model may focus more on alignment loss and overlook diffusion loss, thereby destroying some of the original generation behavior. On the contrary, maintaining $\alpha$ in a very large value (0.9999) ensures the alignment target is extremely robust and slow to change, preventing the alignment loss from destabilizing the primary diffusion loss. Thus, we set the momentum coefficient as 0.9999 as our default.

## G    PRINCIPLE OF OUR HYPERPARAMETER

Although our method requires some hyperparameter tuning, we believe that some settings are model-independent, while some settings have selection principles. We now give the detailed explanations, and we hope these explanations can help to develop SRA to other new models more easily.

- **Projection Head.** In generative models, different layers are often believed to have different generation responsibilities. We hypothesize that using the lightweight projection head instead of directly conducting the alignment can lead to a relatively soft alignment, and could be the reason for avoiding disruption to the model's original generative behavior. We believe this principle and findings with the design of projection head can be easily transferred to new models.

- $\alpha$ **in EMA decay.** In EMA, a larger $\alpha$ indicates a smoother update. We analyze that the reason for keeping such a very large value of $\alpha$ works well in SRA is that: in our method, the enhanced representation learning is ultimately for the service of generation. Thus, when the update of teacher model is relatively intense, the alignment would be less stable, the model may focus more on alignment loss and overlook diffusion loss, thereby destroying some of the original generation behavior. On the contrary, maintaining $\alpha$ in a very large value (0.9999) ensures the alignment target is extremely robust and slow to change, preventing the alignment loss from destabilizing the primary diffusion loss. This can also be indirectly verified by our following analysis of $\lambda$. Given this analysis, we believe the choice of $\alpha$ is will be well-matched to most of the new model.

- **Block layers and time intervals.** Principle of the selection of block layers: the block layer for the teacher is selected so that the corresponding representation has strong semantics (See Figure 2 (b)). The block layer for students is the first few layer as the first few layers are more about semantics learning, which is similar to REPA, and discussed in REPA. Principle of the time interval: The intuitive guidance could be that the target representation for the teacher timestep has better semantics, and the teacher timestep is not too far from the current timestep, i.e., a balance between teacher representation semantics and the semantic distance between teacher (timestep sampled from the interval) and student (current timestep). These principles are also applicable to other diffusion transforms (e.g, MMDiT as shown Table 2).

- **Coefficient $\lambda$.** The first principle we adopt $\lambda$ is to maintain the diffusion loss (approximately 0.7) and the alignment loss (approximately 1.6) in the same scale. Our experimental results indicate that performance is slightly better when the alignment loss is relatively smaller, when these two points are satisfied, we consider this little performance variance is not suspicious. We also conduct additional experiments as follows, suggested that a smaller $\lambda$ (e.g., 0.02) diminishes the regularization effect of the alignment loss, while a larger $\lambda$ (e.g., 2) makes the alignment loss to dominate training instead of assist generation, which can harm sample quality (this can also indirectly verifies why EMA performs well when a large fixed $\alpha$ is set.). Hence, once we know the scale of diffusion loss and the above principles we give, the selection of $\lambda$ will be very easy to new model.

## H  TRAINING SPEED AND GPU MEMORY USAGE AGAINST BASELINES

Table 7: Memory useage, per-epoch training time comparison. "Param" counts only the additional parameters required by the auxiliary forward pass. These results are tested with total batch size of 256 on a single machine with 8 A100 GPUs.

| Method | Mem (GB) | Time (h) | Param |
|---|---|---|---|
| SiT-B | 12.85 | 0.096 | – |
| SiT-B + SRA | 13.23 | 0.115 | 87 M |
| SiT-L | 25.68 | 0.282 | – |
| SiT-L + SRA | 26.65 | 0.330 | 305 M |
| SiT-XL | 29.08 | 0.394 | – |
| SiT-XL + SRA | 30.24 | 0.476 | 481 M |
| DiT-B | 13.96 | 0.173 | – |
| DiT-B + SRA | 14.78 | 0.209 | 87 M |
| DiT-L | 27.85 | 0.503 | – |
| DiT-L + SRA | 29.26 | 0.558 | 305 M |
| DiT-XL | 30.68 | 0.719 | – |
| DiT-XL + SRA | 32.31 | 0.804 | 481 M |

As our approach needs one more forward pass on the diffusion backbone to compute the target representation. We now provide a specific comparison on GPU memory usage and training time with

baselines. Noting that the extra pass is only for forward, and no gradient computation is needed, so we can use half-precision for fast forward. In addition, it is not needed to pass the whole network, and only a partial pass is needed: 8/12 blocks in SiT-B, 20/28 blocks in SiT-XL. So it can be seen that the additional cost are not very large, and the results in Figure 4 and Table 3 also indicate that our method can improve the performance of the model within the same training time.

## I    DETAILED SETUP OF ABLATION STUDY

We now give the detailed experimental setup of Figure 7c in our ablation study. The accuracy rate of the horizontal axis in the figure is obtained by using the linear probing results of the original SiT-XL/2 checkpoint training for 7M iterations, while the vertical axis is the FID evaluation result of training 400K iterations with SRA without classifier-free guidance (CFG). Since we do not introduce any representation component and use the representation supervision signal only in the generative training process, we consider this experiment to validate the effectiveness of our approach.

## J    ABLATION RESULTS OF DIT

We also perform a similar analysis with DiT like those have done in Figure 7c, the results are showed in Figure 9. In short, we also observe that the generative capability of DiT with SRA is indeed strongly correlated with the representation guidance as observed in SiT with SRA.

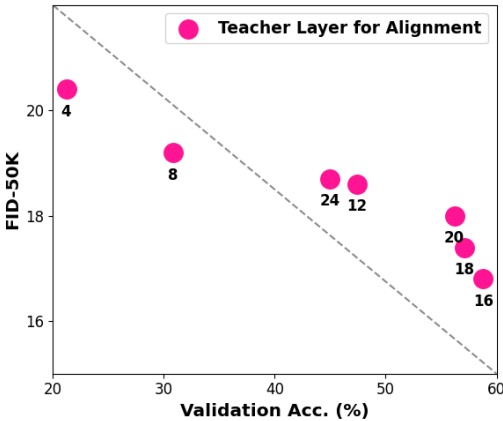

Figure 9: We also investigate the correlation between generation performance and the representation guidance of DiT + SRA. A similar tight coupling can also be seen.

## K    MORE DISCUSSION ON RELATED WORK

We now provide a detailed literature review of other related work.

**EMA model as teacher.** Unlike traditional knowledge distillation (Gou et al., 2021) that uses a well pre-trained model as the teacher, using EMA model (Hunter, 1986) as the teacher can be seen as self-distillation because the weight of the teacher model is obtained by the weighted moving average of the student. This method often needs a feasible pretext task to succeed. For example, MoCo (He et al., 2020; Chen et al., 2021) sets the teacher's output as a momentum queue and uses contrastive learning to guide the student model; DINO (Caron et al., 2021; Oquab et al., 2023) feeds two views of images to the teacher, and the trainable student then forces the student's output distribution to be close to that of the teacher. Our work also shares some similarities, where we set aligning the student model's latent feature in the earlier layer conditioned on higher noise with that in the later layer conditioned on lower noise of the teacher as our pretext task for training.

**Diffusion transformers.** Currently, the diffusion model is progressively transitioning from a U-Net-based architecture to a Transformer-based one, owing to the superior scalability of the latter. At the beginning, U-ViT (Bao et al., 2023) shows transformer-based backbones with *skip connections* can

be an effective backbone for training diffusion models. Then, DiT (Peebles & Xie, 2023) shows skip connections are not even necessary components, and a pure transformer architecture can be a scalable architecture for training denoise-based models. Based on DiT, SiT (Ma et al., 2024) shows the model can be further improved with continuous stochastic interpolants (Albergo & Vanden-Eijnden, 2023). Moreover, Stable diffusion 3 (Esser et al., 2024), Z-Image (Team et al., 2025) and FLUX 1 (Labs, 2024) show pure transformers can be scaled up for challenging text-to-image generation, and this characteristic is also verified by Sora (Brooks et al., 2024), CogvideoX (Yang et al., 2024) and Wan (WanTeam, 2025) in text-to-video field. Our work focuses on improving the training of DiT (and SiT) architecture based on our proposed self-representation alignment technique.

**Exploring representation capacity of diffusion models.** With the success of diffusion models to generate detailed images, many works have attempted to test whether discriminative semantic information can be found in diffusion models. GD (Mukhopadhyay et al., 2023) and DDAE (Xiang et al., 2023) first observe that the intermediate representations of diffusion models have discriminative properties. Driving from this finding, I-DAE (Chen et al., 2024b) deconstructs diffusion models to be a self-supervised Learner. Moreover, Repfusion (Yang & Wang, 2023), DiffusionDet (Chen et al., 2023), and DreamTeacher (Li et al., 2023) use diffusion models to perform various downstream tasks (*e.g.*, semantic segmentation and object detection). Our work also tries to explore the representation capacity of the diffusion model, but we focus on leveraging the representations in diffusion transformers to enhance their generation capacity.

**Applying representation guidance to other generation tasks.** In addition to pre-training of the class-conditional diffusion model, applying representation guidance can benefit other generation tasks. For example, lbGen (Jiang et al., 2025) utilizes text features from CLIP (Radford et al., 2021) as a low-biased reference to regulate diffusion model for low-biased dataset synthetics. RCG (Li et al., 2024) focuses on unconditional generation, which first use the features from a self-supervised image encoder to serve as the 'label' for image generation by a second generator. Diff-AE (Preechakul et al., 2022) uses a learnable encoder for discovering the high-level semantics and a diffusion model for modeling stochastic; this dual-encoding improves the realism of the generated images on attribute manipulation and image interpolation tasks. Different from these works, our study focuses on the class-conditional diffusion transformer's pre-training and exploiting representation guidance in itself and its own training paradigm. But we also believe our plug-and-play method can be easily applied to other tasks with benefits.

**Knowledge distillation for diffusion model.** Knowledge distillation (Gou et al., 2021) is also widely used in diffusion models. Its purposes can roughly be divided into two types: improving generation performance and accelerating inference sampling. To achieve the first goal, A more powerful pretrained model is often used to guide the diffusion model during training (Daniel Verdú, 2024; Yu et al., 2025; Fang et al., 2024). For example, TinyFusion (Fang et al., 2024) combines progressive distillation with architecture-specific optimizations for U-Net backbones, enabling deployment on edge devices. REPA (Yu et al., 2025) distills the knowledge from a large-scale, pretrained representation foundation model (Oquab et al., 2023; Radford et al., 2021) and finds that this distillation can improve both training efficiency and generation quality. Meanwhile, to achieve the second goal, the student and teacher models will be initialized by pre-trained models (here it can be one model or two different models), then the training target is to make the prediction of the student model with fewer steps conform to that of the teacher model with multiple steps (Luo et al., 2023; Ren et al., 2024; Meng et al., 2023; Song et al., 2023), thereby achieving the result of accelerated sampling and generation speed. For example, CM (Song et al., 2023) constrains the student's outputs of adjacent points on a sampling path to be consistent with teacher's. LCM (Luo et al., 2023) leverages this idea in the latent space. Our study also have some similarities, while we do not need a pretrained model and improve the generation performance of diffusion transformers by applying proposed SRA.

## L MORE QUALITATIVE RESULTS (CFG 4.0)

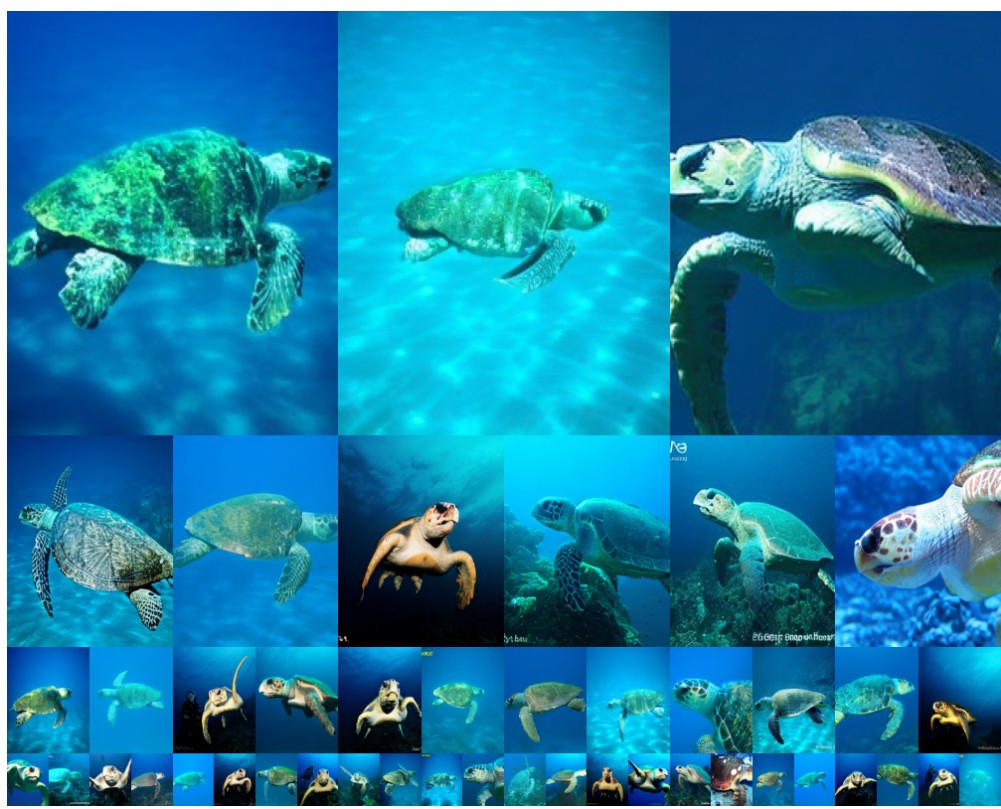

Figure 10: **Uncurated samples of loggerhead turtle (class label: 33).**

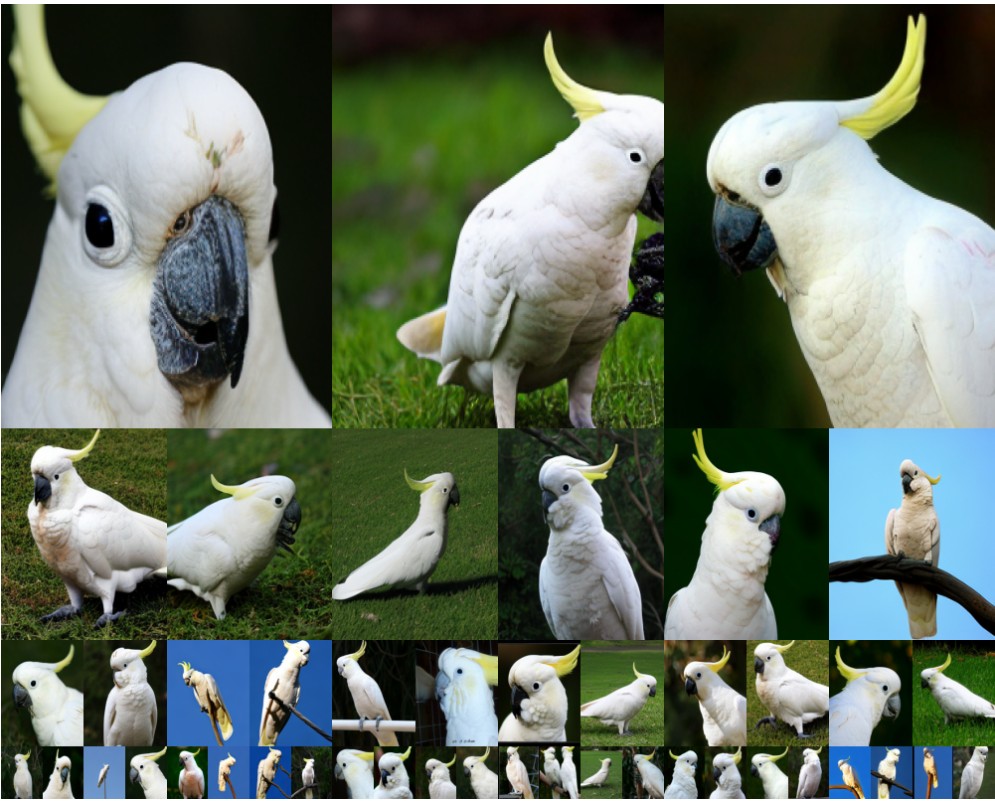

Figure 11: **Uncurated samples of sulphur-crested cockatoo (class label: 89).**

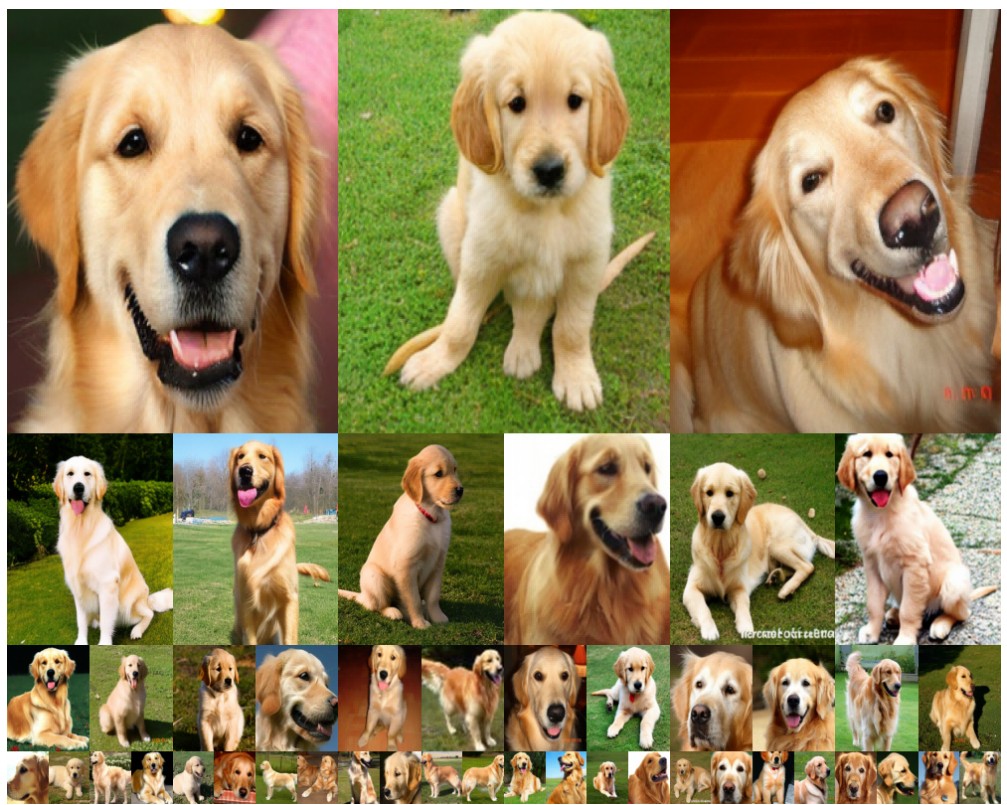

Figure 12: **Uncurated samples of golden retriever (class label: 207).**

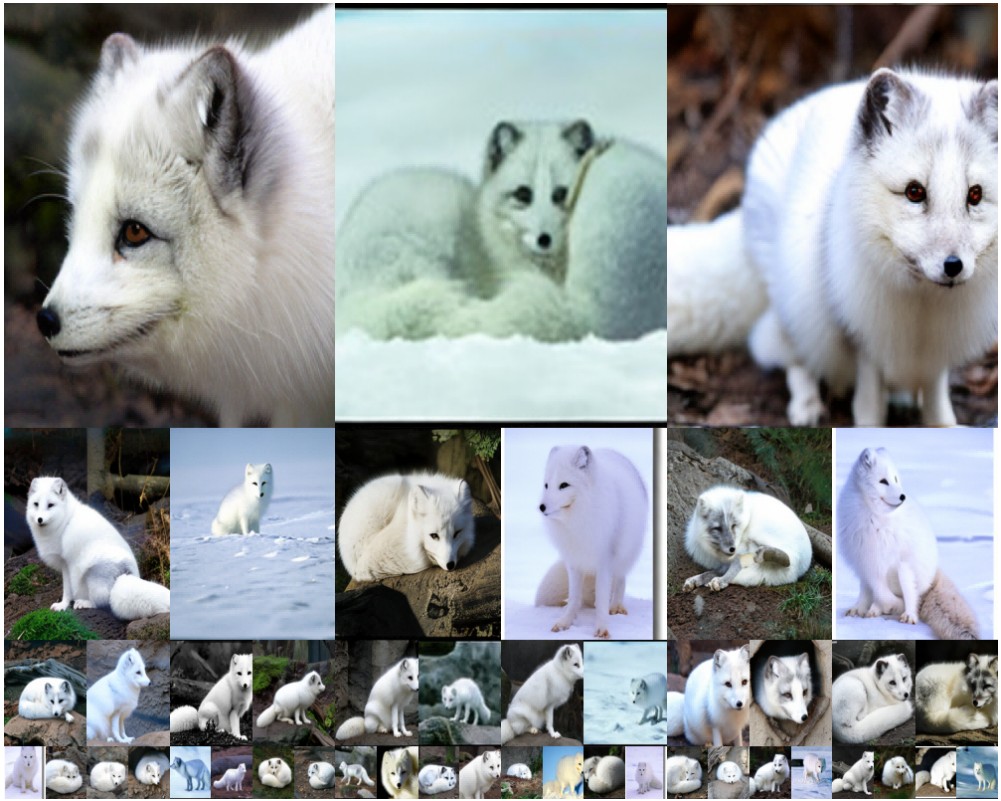

Figure 13: **Uncurated samples of white fox (class label: 279).**

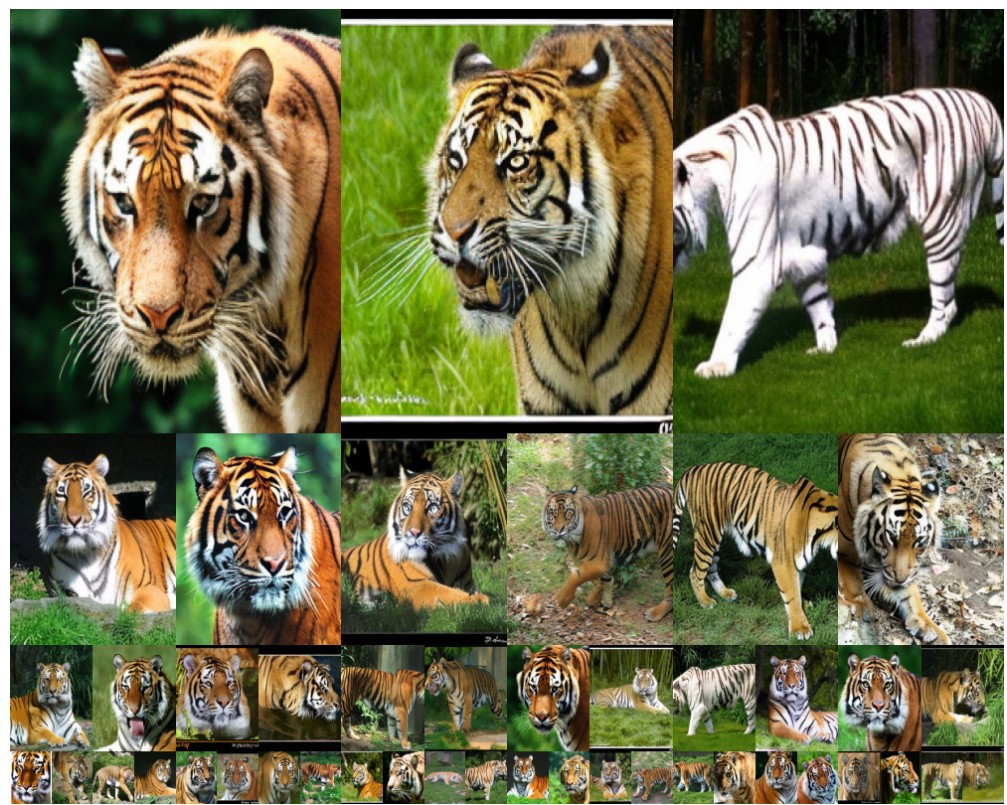

Figure 14: **Uncurated samples of tiger (class label: 292).**

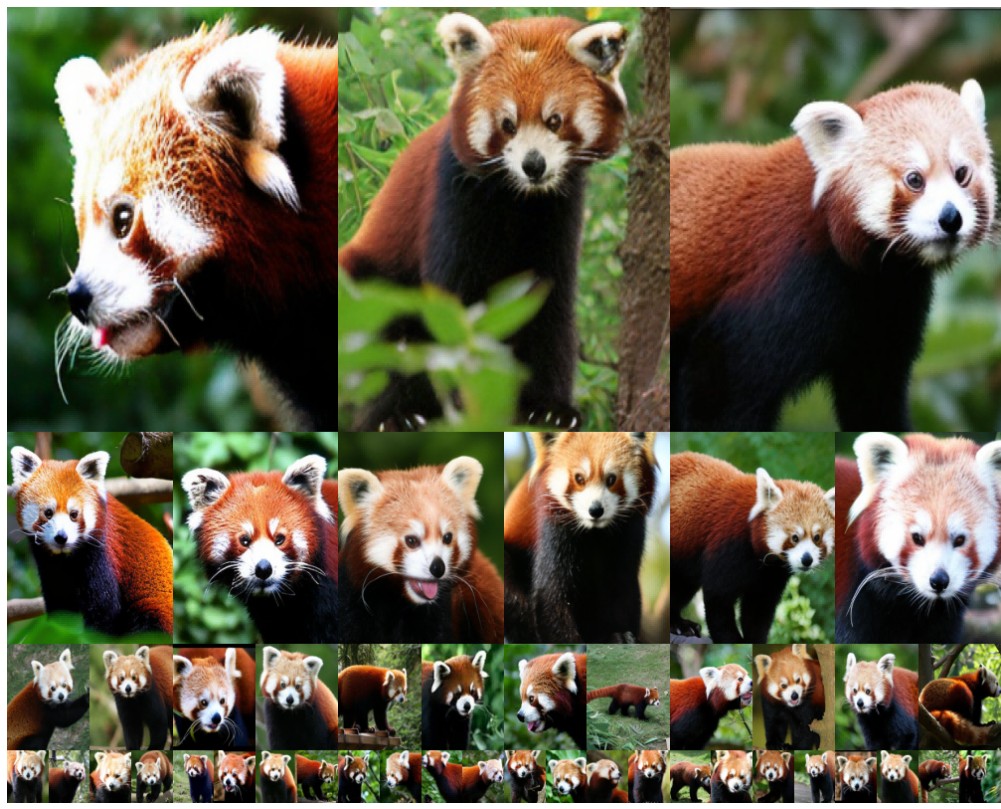

Figure 15: **Uncurated samples of red panda (class label: 387).**

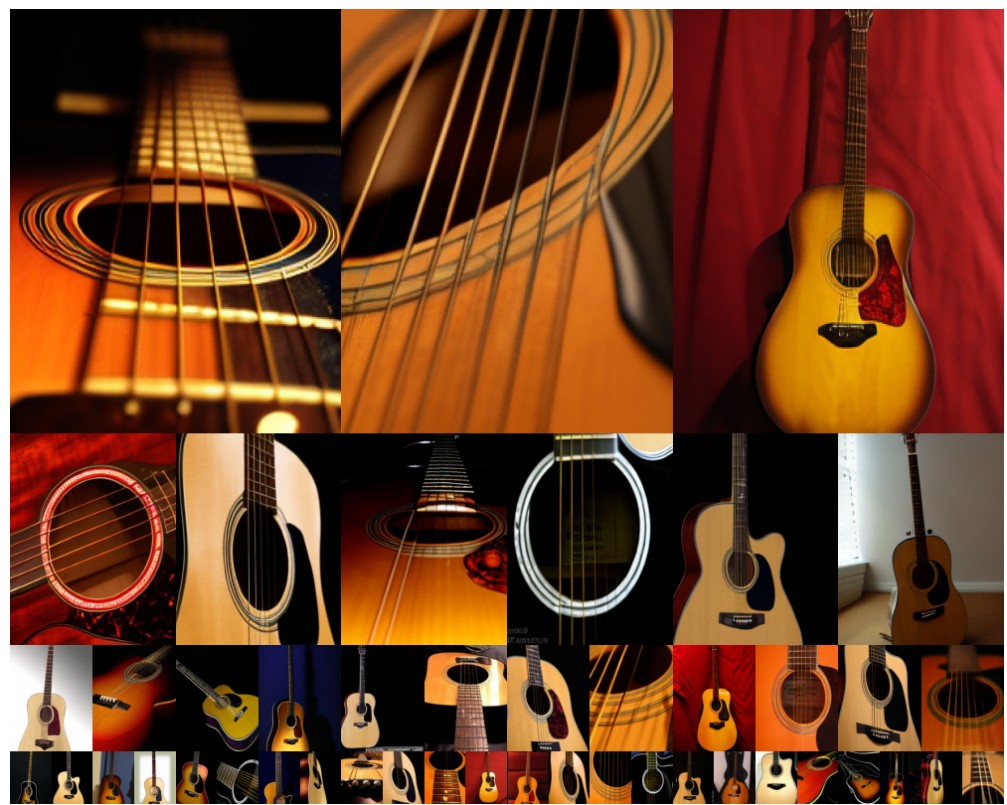

Figure 16: **Uncurated samples of acoustic guitar (class label: 402).**

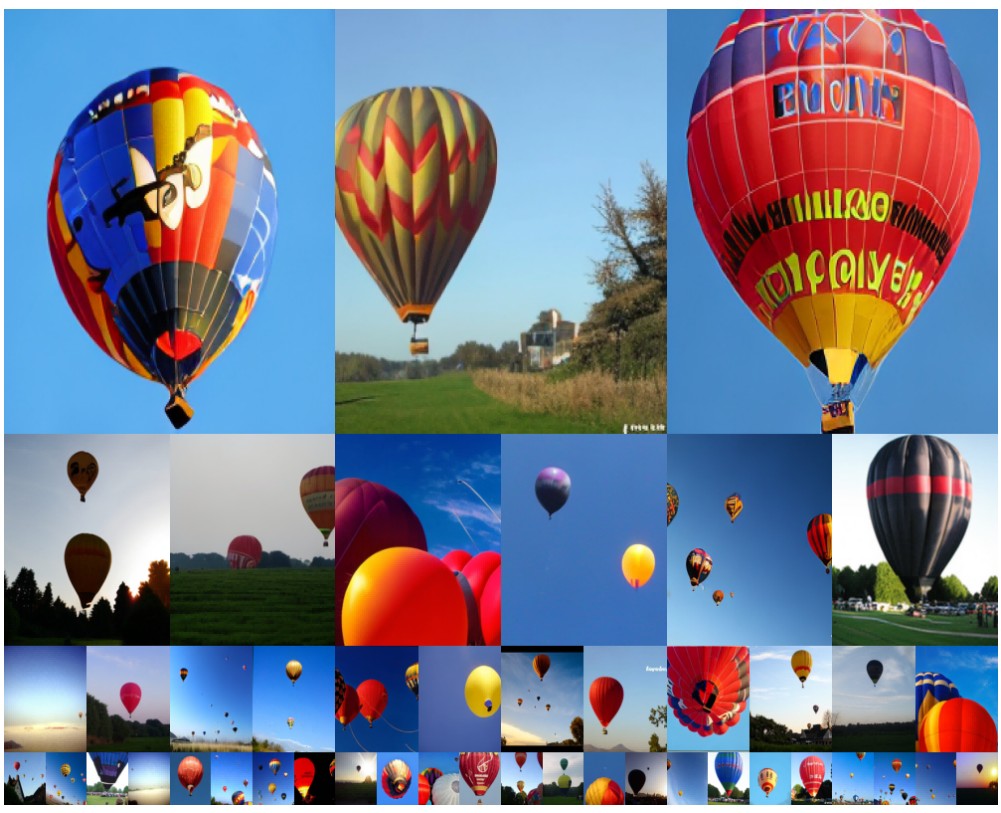

Figure 17: **Uncurated samples of balloon (class label: 417).**

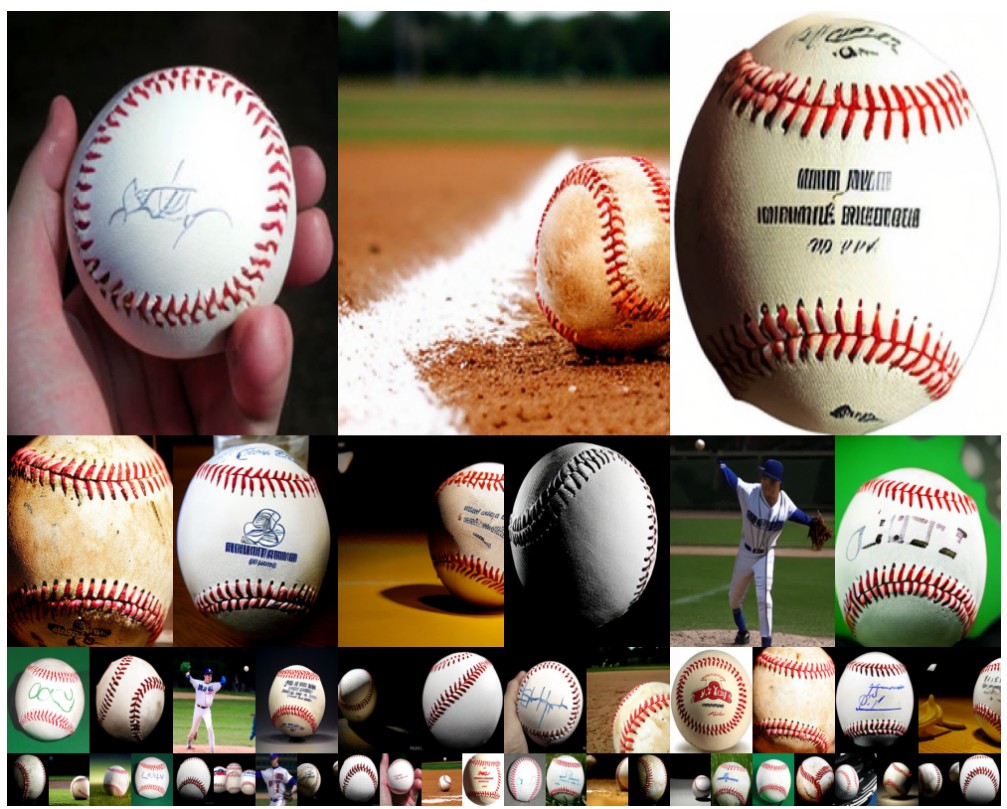

Figure 18: **Uncurated samples of baseball (class label: 429).**

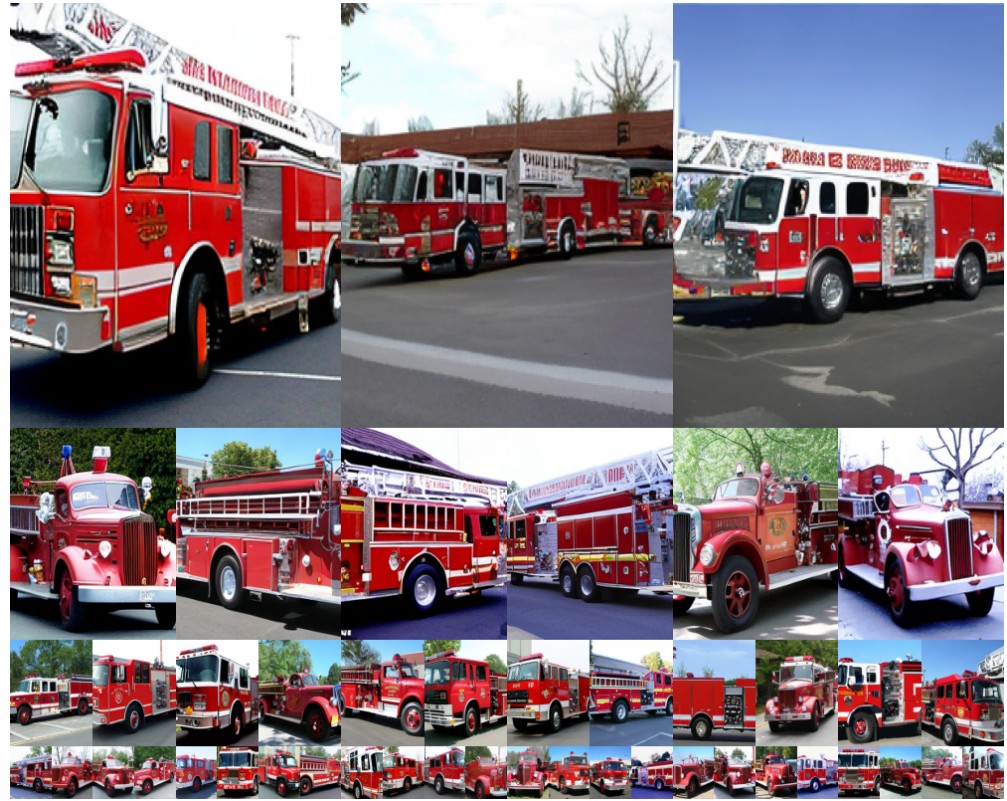

Figure 19: **Uncurated samples of fire truck (class label: 555).**

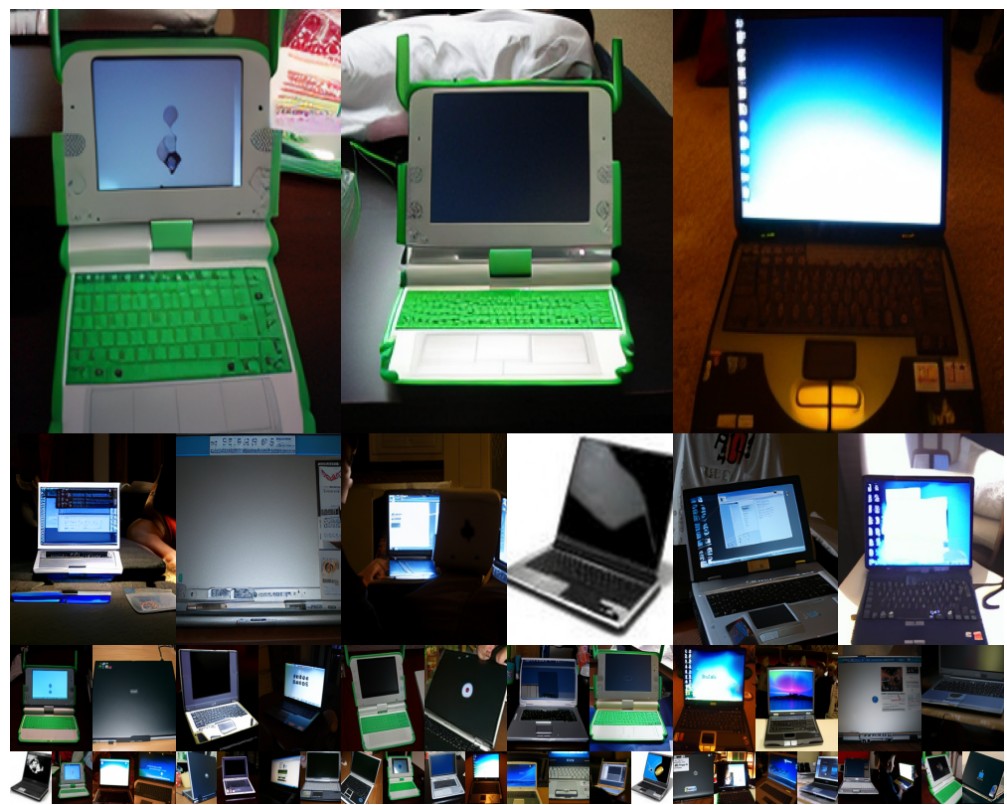

Figure 20: **Uncurated samples of laptop (class label: 620).**

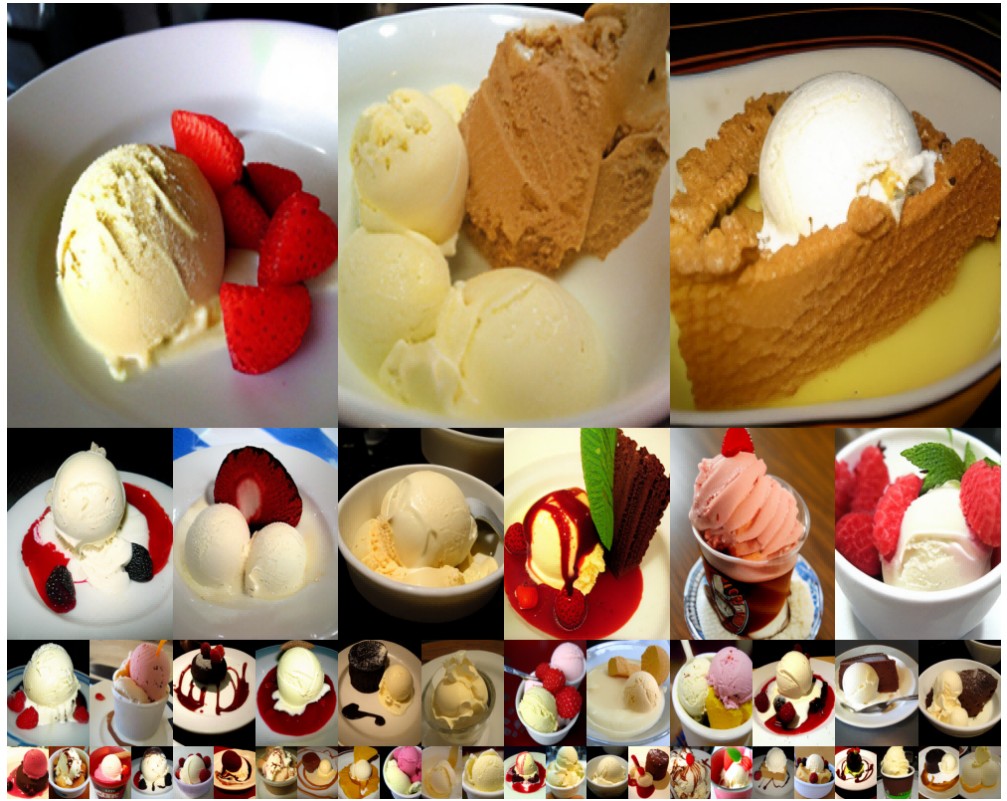

Figure 21: **Uncurated samples of ice cream (class label: 928).**

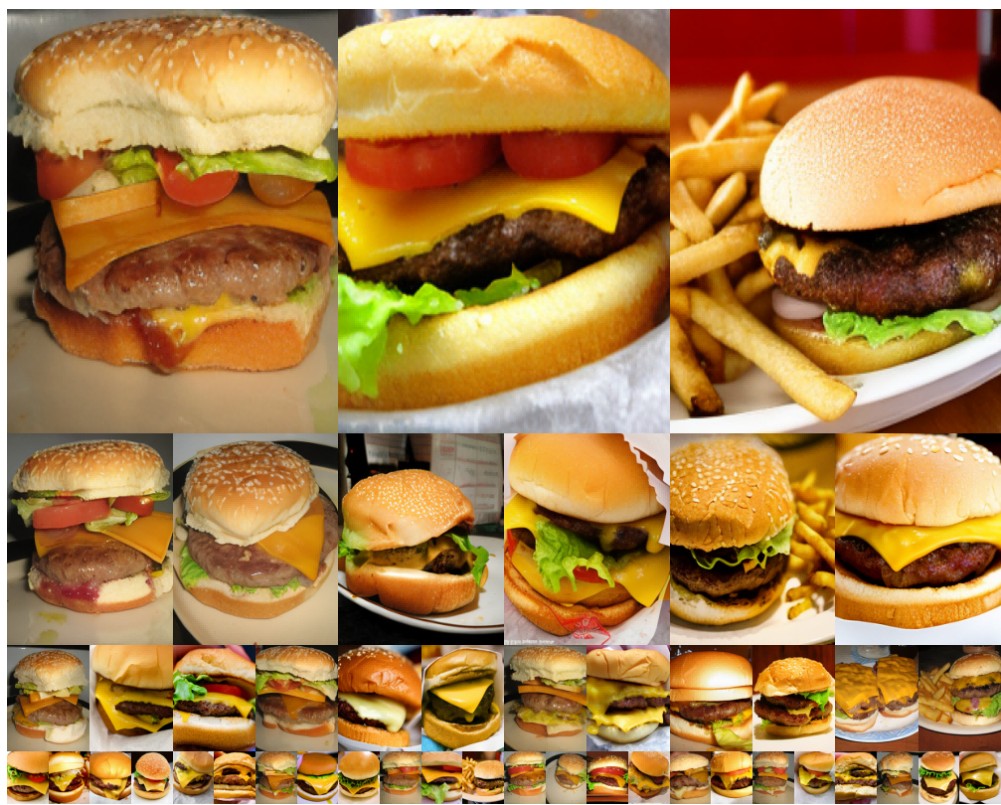

Figure 22: **Uncurated samples of cheeseburger (class label: 933).**

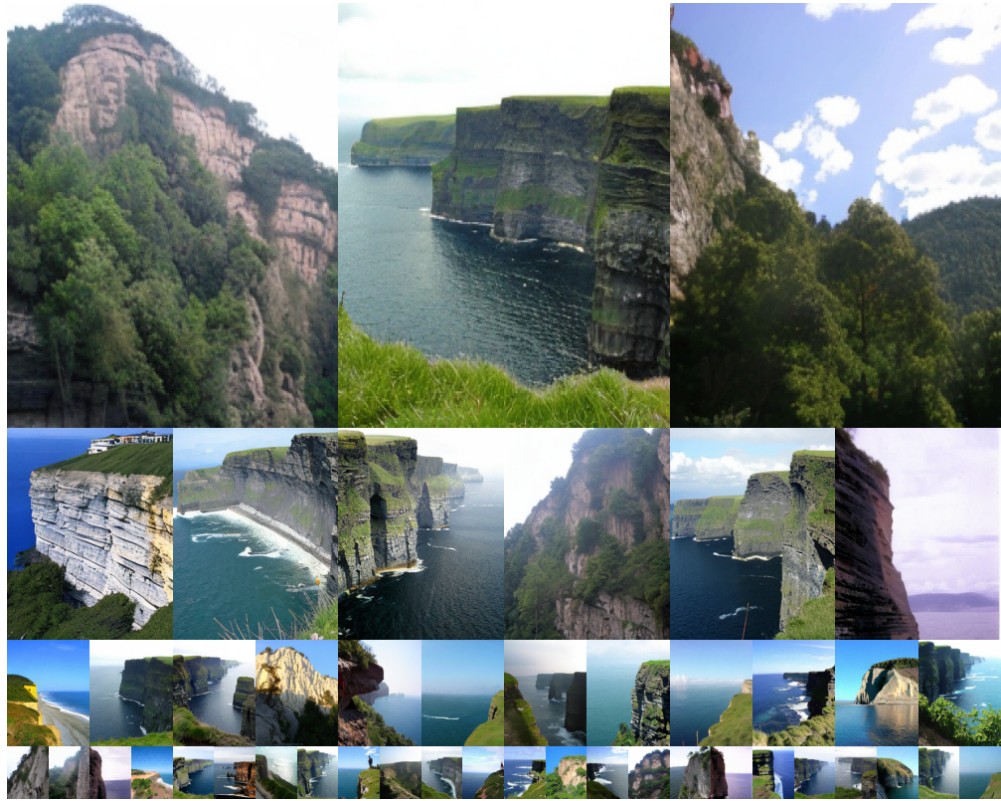

Figure 23: **Uncurated samples of cliff drop-off (class label: 972).**

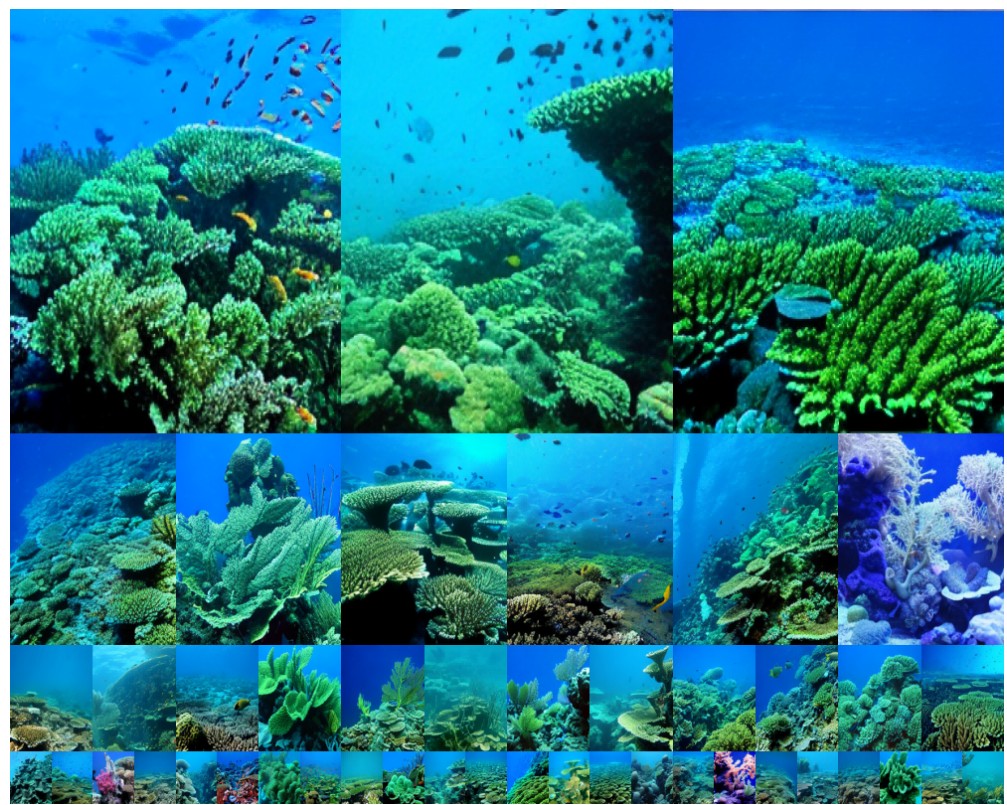

Figure 24: **Uncurated samples of coral reef (class label: 973).**

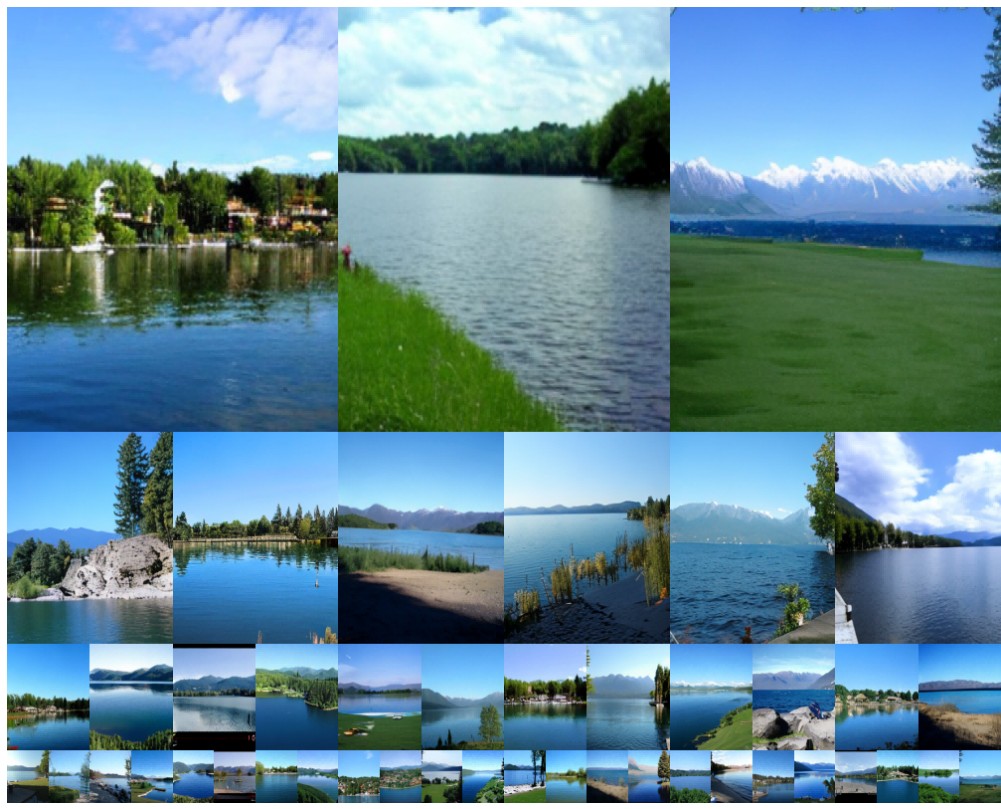

Figure 25: **Uncurated samples of lakeside (class label: 975).**

