# OpenReview forum: "Representation Alignment for Diffusion Transformers without External Components"
_ICLR.cc/2026/Conference — ICLR 2026 Poster_

### Official Review · Reviewer_MZU4 · 2025-10-21

**Soundness:** 2
**Presentation:** 3
**Contribution:** 2
**Rating:** 6
**Confidence:** 3

**Summary:**

This paper introduces Self-Representation Alignment (SRA), which improves representation learning quality in diffusion/flow transformers (DiTs/SiTs) without relying on pre-trained encoders or auxiliary representation learning tasks. SRA achieves accelerated training speed and improved generation performance in practice. The key insight is that diffusion/flow transformers learn a hierarchy of representations which progress from coarse features (in earlier layers and with higher noise levels) to fine features (in later layers and with lower noise levels). SRA leverages this internal structure in a student-teacher framework, where the teacher is an EMA version of the student network. The diffusion/flow training objective is then augmented with an alignment loss that encourages the projection of the student's coarse representations to match the teacher's more refined representations. Large-scale experiments are performed to evaluate SRA's empirical performance.

**Strengths:**

1. The observation of the hierarchy of the internal representations in diffusion/flow transformers is interesting, which is a natural motivation for the proposed method.
2. The key idea of using the EMA model's representation in later layer and with lower noise level for self-guidance is novel. This has the advantage of eliminating the need of pretrained encoder or auxiliary representation learning tasks for representation alignment.
3. Thorough empirical evaluation and ablation studies are performed to demonstrate the proposed method's competitive empirical performance.
4. The paper is easy to understand with a clear description of the preliminaries and the proposed method, as well as a well-organized section for experiment result.

**Weaknesses:**

1. The primary weakness is a lack of a principled foundation, as many core design choices feel arbitrary and are determined through empirical trial-and-error rather than guided by a clear theoretical or analytical framework. Below are some examples:

    (a) The projection head is a bit counterintuitive but is shown to be crucial during training with no clear justification. Intuitively, it feels more natural to perform alignment without projection or with projection for both teacher and student outputs.

    (b) It is unclear why a fixed EMA decay of a particular value works well and whether this is still the optimal value for new tasks/architectures.

    (c) The choices of the layers and time intervals for self-guidance feel ad-hoc.

    (d) It is suspicious that there is only a little performance variance for different values of the alignment loss's weight $\lambda$, given that it controls the trade-off between the diffusion loss and alignment loss.

2. The proposed method requires tuning many hyperparameters and its performance seems highly sensitive to these specific hyperparameter choices, for which the paper offers no clear scientific explanation or general rule. As a result, it is unclear whether these hyperparameter will generalize to new architectures/tasks/settings.

3. Although the authors propose several hypotheses to explain why the chosen hyperparameters make sense, it still does not provide sufficient scientific intuition for why the proposed alignment is useful as the hypotheses are unverified.

4. Another major weakness is that there is no discussion of the limitations of the proposed method in the paper.

5. There are some minor typos in the paper:

    (a) "Dose" -> "Does" in Line 227,

    (b) "Tabel 2" and "Tabel 3" -> "Table 2" and "Table 3" in Line 373,

    (c) "Defualt" -> "Default" in Line 918,

    (d) "useage" -> "usage" in Line 1068.

**Questions:**

1. Could the authors comment on the generalizability of the chosen hyperparameters to other settings/tasks and provide some scientific explanations and/or general rules for selecting these hyperparameters?
2. Could the authors discuss the limitations of the proposed method in the paper?

---

> ### Author Response · Authors · 2025-11-20
> **Response to Reviewer MZU4 (1/2)**
>
> Thanks very much for your thoughtful feedback and the time you devoted to reviewing our manuscript. We respond to each of your comments one by one in what follows:
>
> ---
> **[W1&2&3 and Q1].  Principled rules and explanations of our design choices and test them on text-to-image settings (part1)**
>
>
>
> Thanks for pointing all these out.    As suggestion, we now first give our  principled rules and explanations and then give the illustration of the generalizability of our hyperparameter settings.
>
>
> Indeed,  our method requires some hyperparameter tuning.   But we believe that some settings are model-independent, while some settings have selection principles. We now give the reasons:
>
> (1). Projection Head.     In generative models, different layers are often believed to have different generation responsibilities.  We hypothesize that using the lightweight projection head instead of directly conducting the alignment can lead to a relatively soft alignment, and could be the reason for avoiding disruption to the model’s original generative behavior.   Moreover, following your suggestion, we use projection head for both teacher and student, the results show that this will bring slightly better results compared to our default settings, but the gain is saturated, suggesting a lightweight head  for student is already effective enough. We believe this principle and findings with  the  design of  projection head  can be easily transferred to new models.
>
> | projection head  | FID↓  | IS↑   |
> |------------|-------|-------|
> | No |34.23| 41.07 |
> | student only  | 29.10 | 50.20 |
> | both  | 28.87 | 50.55 |
>
>
>
> (2). α in  EMA decay.  In EMA, a larger  α  indicates a smoother update. We analyze that the reason for keeping such a very large value of α works well in SRA is that: in our method, the enhanced representation learning is ultimately for the service of generation. Thus, when the update of teacher model is relatively intense, the alignment  would be less stable, the model may focus more on alignment loss and overlook diffusion loss, thereby destroying some of the original generation  behavior. On the contrary, maintaining α in a very large value (0.9999) ensures the alignment target is extremely robust and slow to change,  preventing the alignment loss from destabilizing the primary diffusion loss.   This can also be indirectly verified by our following analysis of λ.  Given this analysis and the similar observation in our ablations on t2i task using MMDiT [1] shown below, we believe the choice of α is will be well-matched to most of the new model.
>
>
>  | Method | α | FID↓  | PickScore↑   |
> |------------|--|-------|-------|
> | MMDiT | - | 5.86|  20.05|
> | MMDiT + SRA |0|6.22|  19.93|
> | MMDiT + SRA |0.996 → 1.0|5.58|  20.62|
> | MMDiT + SRA | 0.9999|4.85 | 21.14 |
>
>
> (3).  Block layers and time intervals.  Principle of the selection of block layers: the block layer for the teacher is selected so that the corresponding representation has strong semantics (See Figure 2 (b)). The block layer for students is the first few layer as the first few layers are more about semantics learning, which is similar to REPA, and discussed in REPA.  Principle of the time interval: The intuitive guidance could be that the target representation for the teacher timestep has better semantics, and the teacher timestep is not too far from the current timestep, i.e., a balance between teacher representation semantics and the semantic distance between teacher (timestep sampled from the interval) and student (current timestep). These principles are applicable to other diffusion transforms (e.g, MMDiT as shown latter), so we politely do not agree this is ad-hoc.
>
> (4). Coefficient λ.  The first principle we adopt λ is to maintain the diffusion loss (approximately 0.7) and the alignment loss (approximately 1.6) in the same scale. Our experimental results indicate that performance is slightly better  when the alignment loss is relatively smaller, when these two points are satisfied, we consider this little performance variance is not suspicious. We also conduct additional experiments as follows,  suggested that a smaller λ (e.g., 0.02) diminishes the regularization effect of the alignment loss, while a larger λ (e.g., 2) makes the alignment loss to dominate training instead of assist generation, which can harm sample quality (this can also indirectly verifies why EMA performs well when a large fixed α is set.).  Hence,  once we know the scale of diffusion loss and the above principles we give, the selection of λ will be very easy to new model.
>
>
> | λ   | FID↓  | IS↑   |
> |-----|-------|-------|
> | 0.02 | 32.48 | 44.21 |
> | 0.2 | 29.10 | 50.20|
> | 2 | 40.15 | 35.63 |

---

> ### Author Response · Authors · 2025-11-20
> **Response to Reviewer MZU4 (2/2)**
>
> ---
> **[W1&2&3 and Q1].  Principled rules and explanations of our design choices and test them on text-to-image settings (part2)**
>
> To further test this principle,  we follow REPA to conduct text-to-image generation using MMDiT backbone on MS-COCO[2]. In a specific, we use the 28 MMDiT  blocks (make the number of block layers consistent with that of SiT-XL ) and apply our SRA loss on the double-stream features.
> It is worth noting that we use the same hyperparameter settings as we use in SiTXL + SRA (same block layers for alignment, same time interval, same projection head design, same α in EMA, same λ for alignment) The results below show that SRA can naturally extend to text-to-image generation. The same self-alignment strategy and block layer for alignment works, as evidenced by our experiments: applying SRA to MMDiT without any hyperparameter tuning improves FID (5.66 → 4.75) and PickScore (20.65 → 21.14) over the baseline, and is comparable to REPA.
>
>  | Method | FID↓  | PickScore↑   |
> |------------|-------|-------|
> | MMDiT |5.86|  20.05|
> | MMDiT + REPA  | 4.60 | 20.88 |
> | MMDiT + SRA | 4.85 | 21.14 |
>
>
> *text-to-image results (ODE, NFE=50, 150K iter)*
>
> [1] Scaling Rectified Flow Transformers for High-Resolution Image Synthesis, ICML 2024
>
> [2] Microsoft COCO: Common Objects in Context, ECCV 2014
>
> **[W4 and Q2]. Discussion on limitations of our work**
>
>
>
> Thanks for the suggestion. Due to the limitations of computation resources, we do not validate our method in the scenario of large-scale text-to-video generation  where currently there is no well-pretrained strong encoder for open-domain videos (there is some video pretrained model, e.g., VideoMAE[1], which is still not strong enough for open-domain video encoder, compared to DINO for image encoder), but we have already demonstrated the potential of our method on ImageNet ( performance comparable to that of REPA which use strong DINOv2 for supervision). Applying our method to video generation would be an interesting direction.  Moreover, similar to other related works like REPA,  our method is also experiment-driven. Exploring theoretical insights into why learning a good representation is beneficial to generation will also be an exciting future direction. We also added such discussions in Section 6 in the revised manuscript.
>
> [1] VideoMAE V2: Scaling Video Masked Autoencoders with Dual Masking, CVPR 2023
>
>   ---
>  **[W5]  Typos.**
>
>  Thanks for the correction. We have corrected it in the latest submitted version

---

### Official Review · Reviewer_1H2k · 2025-10-27

**Soundness:** 3
**Presentation:** 3
**Contribution:** 2
**Rating:** 4
**Confidence:** 4

**Summary:**

This paper aims to accelerate the training of diffusion transformers by learning meaningful internal representations while training the model. In particular, this paper proposes SRA, which align the diffusion model with a better representation without external guidance. The key idea is to introduce a teacher network (EMA), and then use different noise and layers of the teacher network for alignment to self-distill a better representation. Experiments demonstrate that SRA accelerates generative training without an external model.

**Strengths:**

1. The paper is overall easy to follow, and the method is simple yet effective.

2. Using self-distillation from the teacher network to boost the training of diffusion models is an intriguing idea.

2. SRA shows faster training with respect to time compared to vanilla Diffusion Transformers, showing the effectiveness of the method.

**Weaknesses:**

1. The authors claim that later blocks with small noise have fine-grained features compared to the first blocks with large noise. However, it does not indicate that later blocks with small noise have better representations to be aligned. For instance, an extremely larger layer even has more detailed features, but it is not beneficial (e.g., 4 -> 12 performs worse than naive SiT in Table 1). Therefore, I think it needs additional / more analysis to justify the effectiveness of the proposed method.

2. The choice of block layers is quite sensitive. In particular, it sometimes performs worse than vanilla SiT. It (1) makes the practitioners to difficult to use this technique to improve the training speed of Diffusion Transformers, and (2) makes it difficult to extend the suggested framework to different domains.

**Questions:**

Please answer the weaknesses.

---

> ### Author Response · Authors · 2025-11-20
> **Response to Reviewer 1H2k (1/1)**
>
> Thanks very much for your thoughtful feedback and the time you devoted to reviewing our manuscript. We respond to each of your comments one by one in what follows:
>
> ---
>  **[W1] Correct expression to avoid confusion about the features trend.**
>
>   Thanks for pointing it out.  The "coarse-to-fine" in our paper means "bad-to-good" not "global to  fine-grained".  And our motivation is to find better representations in the diffusion transformer itself to guide the weaker representations, thereby improving its overall representation learning.  In our observation and analysis, the the deeper  block layers  with small noise  have better representations than the earlier layers with higher noise as evidenced in Figure 2 (b), thus makes it  better to serve as alignment target. Moreover, we find that the representation dropps sharply in the extremely later layer (after 20 for SiT-XL shown in in Figure 2 (b)), and the Figure 7 (c) shows a strong correlation between students's generation quality and the teacher's representation quality in our method. This why 4 -> 12 performs worse than naive SiT in Table 1 (12 is the last layer in SiT-B).  We have changed all  "coarse-to-fine"  to  "bad-to-good"  in our latest submission version to avoid confusion.
>
>
>   ---
>   **[W2]. Principle of our  hyperparameter settings about block layers and test it on text-to-image settings**
>
>   This is a good point.  Although our method indeed requires the hyperparameter tuning of block layers, we believe that the  principle we give  below and findings in our paper is  model-agnostic and can reduce the search space:  The block layer for the teacher is selected so that the corresponding representation has strong semantics (in deeper but not the last layers). The block layer for students is the first few layer as the first few layers are more about semantics learning, which is similar to REPA, and discussed in REPA.   Thus, when such a principle is in place, choosing hyperparameters will be much easier. (e.g, 3 -> 8, 2-> 8, 4 ->10, etc  demonstrate better performance than the baseline)
>
> To further test this principle,  we follow REPA to conduct text-to-image generation using MMDiT[1] backbone on MS-COCO[2]. In a specific, we use the 28 MMDiT  blocks (make the number of block layers consistent with that of SiT-XL in order to fully reuse the hyperparameter settings  of SRA in SiT-XL.) and apply our SRA loss on the double-stream features.  The results below shows that SRA can naturally extend to text-to-image generation. The same self-alignment strategy and block layer for alignment works, as evidenced by our experiments: applying SRA to MMDiT without any hyperparameter tuning improves FID (5.86 → 4.85) and PickScore (20.05 → 21.14) over the baseline, and is comparable to REPA.
>
>  | Method | FID↓  | PickScore↑   |
> |------------|-------|-------|
> | MMDiT |5.86|  20.05|
> | MMDiT + REPA  | 4.60 | 20.88 |
> | MMDiT + SRA | 4.85 | 21.14 |
>
>
> *text-to-image results (ODE, NFE=50, 150K iter)*
>
> [1] Scaling Rectified Flow Transformers for High-Resolution Image Synthesis, ICML 2024
>
> [2] Microsoft COCO: Common Objects in Context, ECCV 2014

---

> > ### Comment · Reviewer_1H2k · 2025-11-27
> >
> > Thank you for the rebuttal. The replies have addressed my concerns. Thus, I will raise the rating to 6.

---

> > > ### Author Response · Authors · 2025-11-27
> > > **Glad to solve your concerns**
> > >
> > > Dear Reviewer 1H2k,
> > >
> > > We are happy to hear that our rebuttal has successfully solve your problem.  Also, we appreciate your support for our work. If you have any further questions or suggestions, please do not hesitate to let us know.
> > >
> > > Best,
> > >
> > > Authors

---

### Official Review · Reviewer_NzJf · 2025-10-30

**Soundness:** 3
**Presentation:** 3
**Contribution:** 3
**Rating:** 4
**Confidence:** 4

**Summary:**

This paper introduces Self-Representation Alignment (SRA), leveraging internal representation guidance during diffusion transformer training. The core idea is to align latent representations from earlier layers (conditioned on higher noise) with those from later layers (conditioned on lower noise), without relying on any external models. To achieve this, the authors employ an EMA teacher of the same network and apply a lightweight projection head with a patch-wise distance loss. Experimental results show that SRA attains performance comparable to methods using external alignment.

**Strengths:**

- The motivation and flow of ideas in this paper are clear. It starts by analyzing the internal representations of diffusion transformers and then proposes a way to effectively use them to speed up training and improve performance. The paper also shows self-alignment can serve as a good alternative in the absence of external encoders.

- The paper is well-written and easy to follow. The figures are clear and helpful, and the authors include an analysis of training cost, which is one of the considerations with having another network forward pass.

- Experiments across different resolutions (e.g., 512×512) and models (DiTs and SiTs) show that the method scales well and generalizes effectively.

**Weaknesses:**

- The EMA teacher is one of the key components, but the paper gives little discussion about it. From Table 5, the results are quite sensitive to how $\alpha$ is updated - some settings (e.g., 0.0 and 0.996 → 1.0) even show worse FID than the baseline. A deeper analysis beyond saying it is “commonly used” in other works would make this part more convincing.

- The paper does not include experiments or discussions demonstrating SRA’s effectiveness in cases where strong external encoders are unavailable - even though this is the main motivation. It would be helpful to include or discuss results in another setting (e.g., video generation) to make the motivation more convincing.

- The authors claim that “the teacher’s constantly improving capacity allows better representation guidance as training proceeds,” but there is no evidence supporting this. Showing how the teacher’s representation quality (e.g., via linear probing) improves over time would support the claim. Otherwise, the statement should be toned down.

**Questions:**

- Figure 6 shows that SRA keeps improving while external alignment methods saturate early. It gives a good signal, but it’s still unclear whether SRA eventually converges or continues improving over longer training. Could the authors provide longer training curves or convergence analysis?

- I’m also curious whether SRA can naturally extend to text-to-image generation. Would the same self-alignment strategy work there, or are there additional challenges?

- (Minor) Typo in line 373 - “Tabel” → “Table”.

---

> ### Author Response · Authors · 2025-11-20
> **Response to Reviewer NzJf (1/2)**
>
> Thanks very much for your thoughtful feedback and the time you devoted to reviewing our manuscript. We respond to each of your comments one by one in what follows:
>
> ---
>  **[W1] More discussion on α of EMA model**
>
>  Thanks for pointing it out. First, student copy (α = 0.0) performs the worst : this is consistent with the observation in self-supervised learning: student copy leads to unstable training and cannot converge; there is a diffusion loss in our approach, so the training still converges. Second, using α = 0.9999 is better than 0.996 → 1.0:  In EMA, a larger  α  indicates a smoother update. We analyze that the reason for keeping such a very large value of α works well in SRA is that: in our method, the enhanced representation learning is ultimately for the service of generation. Thus, when the update of teacher model is relatively intense, the alignment  would be less stable, the model may focus more on alignment loss and overlook diffusion loss, thereby destroying some of the original generation  behavior. On the contrary, maintaining α in a very large value (0.9999) ensures the alignment target is extremely robust and slow to change,  preventing the alignment loss from destabilizing the primary diffusion loss.
>
>  ---
>  **[W2] Discussions on SRA’s effectiveness in cases where strong external encoders are unavailable.**
>
>  Thanks for pointing these out.  Due to limitations in compuation resources, we are unable to conduct large-scale text-to-video (t2v) pretraining with SRA. However, we maintain that this motivation is conceptually sound for the following reasons. First, even within the image domain (already exist strong encoder like DINOv2), our method achieves performance comparable to that of REPA, as demonstrated in Table 2, Table 3 of the main paper, and Table in Q4, this supports our confidence in the effectiveness of approach in text-to-video generation, where currently there is no well-pretrained strong encoder for open-domain videos (there is some video pretrained model, e.g., VideoMAE[1], which is still not strong enough for open-domain video encoder, compared to DINOv2 for image encoder). Second, in the video domain, numerous studies [2,3] have shown that large-scale pretrained text-to-video (t2v) models already capture rich and transferable representations suitable for various understanding tasks, further reinforcing the potential of SRA in video-related applications. These points have been incorporated into Section 5 of the revised manuscript.
>
>
> [1] VideoMAE V2: Scaling Video Masked Autoencoders with Dual Masking, CVPR 2023
>
> [2]  Video models are zero-shot learners and reasoners, Arxiv 2025
>
> [3] Exploring Pre-trained Text-to-Video Diffusion Models for Referring Video Object Segmentation, ECCV 2024
>
>
>   ---
>  **[W3] Teacher’s representation quality improves over time.**
>
> Thanks for the insightful suggestion. We agree that showing the teacher's improving capacity is crucial. To this end, we use layers of the teacher and the student used for alignment to report their linear probing results.  As shown in the supplementary results below, the teacher's representation quality consistently improves throughout training (from 38.1 at 200K iterations to 54.2 at 800K iterations) and consistently outperforms the student model at each stage. This empirical evidence directly supports our claim that the teacher's improving capacity provides better representation guidance as training proceeds.
>
>   | Iteration |student  | teacher
> |-----|-------|---|
> | 200K | 25.4 | 38.1 |
> | 400K | 39.5 | 49.3|
> | 800K | 47.8 | 54.2 |
>
> *Linear probing accuracy (in %) at different interation*
>
>
>   ---
>  **[W4]  Typos.**
>
>  Thanks for the correction. We have corrected it in the latest submitted version

---

> ### Author Response · Authors · 2025-11-20
> **Response to Reviewer NzJf (2/2)**
>
> ---
>  **[Q1] Longer training**
>
> Thanks for the insightful suggestion. Due to time constraints, we use the official ckpt from REPA and the ckpt of our method in 800 epoch continue training for 150 epoch. The results in the table below show that while REPA's performance saturates, SRA maintains a relatively continuous performance gain.
>
>   | Epoch  | FID (REPA) | FID (SRA)
> |-----|-------|---|
> | 800 | 5.9 | 6.8 |
> | 850 | 5.9 | 6.5|
> | 900 | 5.9 | 6.3 |
> | 950 | 5.8 | 6.1 |
>
>
>
>   ---
>  **[Q2] Text-to-image results with SRA**
>
>  This is a good question. Following your suggestion,  we follow REPA to conduct text-to-image generation using MMDiT[1] backbone on MS-COCO[2]. In a specific, we use the 28 MMDiT  blocks (make the number of block layers consistent with that of SiT-XL) and apply our SRA loss on the double-stream features.  Note that we  fully reuse the hyperparameter settings of SRA in SiT-XL. The results below shows that SRA can naturally extend to text-to-image generation. The same self-alignment strategy works, as evidenced by our experiments: applying SRA to MMDiT without any
>  Hyperparameter tuning improves FID (5.86 → 4.85) and PickScore (20.05 → 21.14) over the baseline, and is comparable to REPA.
>
>  | Method | FID↓  | PickScore↑   |
> |------------|-------|-------|
> | MMDiT |5.86|  20.05|
> | MMDiT + REPA  | 4.60 | 20.88 |
> | MMDiT + SRA | 4.85 | 21.14 |
>
>
> *text-to-image results (ODE, NFE=50, 150K iter)*
>
> [1] Scaling Rectified Flow Transformers for High-Resolution Image Synthesis, ICML 2024
>
> [2] Microsoft COCO: Common Objects in Context, ECCV 2014

---

> > ### Comment · Reviewer_NzJf · 2025-11-27
> >
> > Thank you very much to the authors for the rebuttal with additional experiments and discussions. Most of my concerns are resolved and raise my score to 6.

---

> > > ### Author Response · Authors · 2025-11-27
> > > **Glad to solve your concerns**
> > >
> > > Dear Reviewer NzJf,
> > >
> > > We are happy to hear that our rebuttal has successfully solve most of your concerns. Also, we appreciate your support for our work. If you have any further questions or suggestions, please do not hesitate to let us know.
> > >
> > > Best,
> > >
> > > Authors

---

### Official Review · Reviewer_YcnN · 2025-10-31

**Soundness:** 4
**Presentation:** 4
**Contribution:** 4
**Rating:** 6
**Confidence:** 4

**Summary:**

This paper proposes SRA, a model designed to enhance the generation performance of diffusion transformer models. The key idea of SRA originates from an observation that earlier timesteps and shallow layers of a diffusion transformer capture coarse latent features, while later timesteps and deeper layers perform progressive refinement. Instead of relying on a pre-trained representation encoder (a recent approach for improving diffusion transformer performance) the authors employ an EMA-updated teacher network derived from the training process itself. Through this self-updating training mechanism, the model refines its representations over time. The proposed method is evaluated across various models and achieves performance comparable to those that use large pre-trained encoders.

**Strengths:**

- The paper presents an intuitive and elegant idea: achieving refinement within the model itself during training, without depending on large-scale pre-trained encoders. This demonstrates the potential for applying the method to settings where such encoders are unavailable.

- The proposed method can be easily integrated into existing diffusion training pipelines, and the authors confirm its applicability to both flow-based and diffusion-based models.

- The linear probing results of the learned latent representations provide yet convincing evidence that the latent quality has been improved.

**Weaknesses:**

- The method requires model-specific hyperparameter tuning. For each new model, appropriate block layers need to be identified, and while the time interval and lambda parameters are less sensitive, they still require some search.

- Since the training process involves an EMA teacher model, the approach is likely to increase GPU memory consumption. It would be helpful if the authors could quantify the additional GPU usage and training time per epoch compared to the baseline.

**Questions:**

- Given sufficient GPU capacity, it seems feasible to combine this method with REPA for joint training. Have the authors explored this possibility?

- There seems to be a noticeable performance difference depending on whether the projection head is used or not. Could the authors elaborate on why the projection head contributes so significantly to performance, and what specific role it plays in the model?

---

> ### Author Response · Authors · 2025-11-20
> **Response to Reviewer YcnN (1/2)**
>
> Thanks very much for your thoughtful feedback and the time you devoted to reviewing our manuscript. We respond to each of your comments one by one in what follows:
>
> ---
> **[W1] Principle of our  hyperparameter settings for easier transferring**
>
>
> This is a good point.  Although our method indeed requires some hyperparameter tuning, we believe that some of the principle below depends on the findings in our paper is model-agnostic and can reduce the search space.
>
> (1)  block layers:  the block layer for the teacher is selected so that the corresponding representation has strong semantics (See Figure 2 (b)). The block layer for students is the first few layer as the first few layers are more about semantics learning, which is similar to REPA, and discussed in REPA. Thus, when such a principle is in place, choosing hyperparameters will be much easier. (e.g, 3 -> 8, 2-> 8, 4 ->10, etc demonstrate better performance than the baseline)
>
> (2) time interval:   the target representation in the teacher with  lower noise level; to-be-aligned representation in student  with  higher noise level. And this interval  can not be too large. Once this  principle is in place, the search for hyperparameters will be much easier. (e.g, for flow matching model, t always in [0,1] as SiTs)
>
>
> (3) lambda:  we adopt lambda to maintain the diffusion loss (approximately 0.7) and the alignment loss (approximately 1.6) in the same scale. Our experimental results indicate that performance is slightly better when the alignment loss is relatively smaller, when these two points are satisfied, we consider a little performance variance is acceptable.
>
>
> To further test this principle,  we follow REPA to conduct text-to-image generation using MMDiT[1]  backbone on MS-COCO[2]. In a specific, we use the 28 MMDiT  blocks (make the number of block layers consistent with that of SiT-XL ) and apply our SRA loss on the double-stream features.
> It is worth noting that we use the same hyperparameter settings as we use in SiTXL + SRA (same block layers for alignment, same time interval, same projection head design, same α in EMA, same lambda for alignment) The results below show that SRA can naturally extend to text-to-image generation. The same self-alignment strategy and block layer for alignment works, as evidenced by our experiments: applying SRA to MMDiT without any hyperparameter tuning improves FID (5.66 → 4.75) and PickScore (20.65 → 21.14) over the baseline, and is comparable to REPA.
>
>  | Method | FID↓  | PickScore↑
> |------------|-------|-------|
> | MMDiT |5.86|  20.05|
> | MMDiT + REPA  | 4.60 | 20.88 |
> | MMDiT + SRA | 4.85 | 21.14 |
>
>
> *text-to-image results (ODE, NFE=50, 150K iter)*
>
> [1] Scaling Rectified Flow Transformers for High-Resolution Image Synthesis, ICML 2024
>
> [2] Microsoft COCO: Common Objects in Context, ECCV 2014
>
>  ---
>  **[W2] GPU usage and training time comparison**
>
>  Thanks for point it out. We have provide such quantitative result in the Table 6 in Appendix in the first submission paper (now Table 7). For ease of viewing, we now present the results below and give our analysis.
> In practice, the SiT baseline also needs to update an EMA model,  so the  additional costs only  comes from one additional forward pass of the EMA diffusion network and loss backward. However, it is noted that this extra pass is forward-only and no gradient computation for the teacher is needed, which allows us to use half-precision for a fast and memory-efficient forward. Moreover, the whole network is not involved; only a partial pass is required: 8 of 12 blocks in SiT-B and 20 of 28 blocks in SiT-XL. Hence, as we show in Fig 4 in our main paper,  considerd FID vs. Training Time,  diffusion transformers trained with SRA still demonstrate substantial improvements in performance at the same training time.
>
> | Method      | Mem | Time (A100) |
> |-------------|-----|------|
> | SiT-B       | 12.85 | 0.096 |
> | SiT-B+SRA  | 13.23 | 0.115 |
> | SiT-L      | 25.68  | 0.282 |
> | SiT-L+SRA  | 26.65 | 0.330 |
> | SiT-XL      | 29.08 | 0.394 |
> | SiT-XL+SRA  | 30.24 | 0.476 |
>
> *Batch size 256 , 1 epoch equals to 5K iteration*
>
> *Mem: GPU memory cost (GB)*
>
> *Time: training time per iteration (h/epoch)*

---

> ### Author Response · Authors · 2025-11-20
> **Response to Reviewer YcnN (2/2)**
>
> ---
>  **[Q1] Jointly train with REPA**
>
>
>  This is a good point.  First, we would like to highlight the position of our work: Our method explores how to utilize the internal representation of the diffusion transformers to achieve representation alignment without external components. Removing the dependence on an external encoder makes our approach more general. But as suggested, our method can also be jointly trained with REPA. As shown below, the integration of REPA and SRA offers advantages. Notably, our findings indicate that assigning different projection heads can enhance the effect, suggesting that aligning with different features and implementing appropriate decoupling strategies can further stimulate performance. Future research on additional decoupling and integration strategies (e.g., timestep) would be an interesting direction.
>
>
>  | Method  | projection head | FID↓ |
> |-----|-------|---|
> | Baseline | - | 17.2 |
> | REPA | default | 7.9 |
> | SRA |  default | 11.9|
> | REPA + SRA | shared | 8.8 |
> | REPA + SRA | decoupled  | 7.3 |
>
> *SiT-XL Batch size 256 with 400K iterations*
>
>
>   ---
>  **[Q2]  Role of projection head**
>
>   This is a good point.  In generative models, different layers are often believed to have different generation responsibilities.  We hypothesise that using the lightweight projection head  instead of directly conduct the alignment can lead to a relatively soft alignment, and could be the reason for avoiding disrupting the model’s original generative behavior.

---

> ### Comment · Reviewer_YcnN · 2025-11-26
>
> My concerns have been adequately addressed. (Hyperparameter searching, Gpu usage, Compatibility with REPA)
> Therefore, I will maintain my positive score.

---

> > ### Author Response · Authors · 2025-11-27
> > **Glad to solve your concerns**
> >
> > Dear Reviewer YcnN,
> >
> > We are happy to hear that our rebuttal has successfully solve your problem.  Also, we appreciate your support for our work. If you have any further questions or suggestions, please do not hesitate to let us know.
> >
> > Best,
> >
> > Authors

---

### Author Response · Authors · 2025-11-20
**General Response**

Dear reviewers and AC,

We sincerely appreciate your valuable time and effort spent reviewing our manuscript.

We propose SRA,  improving diffusion transformers training by aligning their own representations in different layers with different noise to achieve representation alignment without external component.


As reviewers highlighted, our method:
- a novel and elegant idea (Reviewer YcnN, NzJf, 1H2k,  MZU4)
- simple, effective and easy to follow.  (Reviewer YcnN, NzJf, 1H2k,  MZU4)
-  comprehensive experiments and analysis (Reviewer YcnN, NzJf,  MZU4)


We appreciate your constructive feedback on our manuscript. In response to your comments, we have meticulously revised and enhanced the manuscript as follows:
- Text-to-image experiment (Section 3.3 and Table 2)
- More hyperparameter setting analysis (Appendix F and Appendix G)
- Additional Linear probing experiment ( Section 3.4 and Figure 7 (d))
- Limitation and Future Work (Section 5)
- Correct expression ("coarse to fine" -> "bad to good")
- Fixing typos ("Dose" -> "Does", "Tabel 2" and "Tabel 3" -> "Table 2" and "Table 3",  "Defualt" -> "Default", "useage" -> "usage")

 In the revised version, these updates are temporarily highlighted in blue for your convenience.

We sincerely believe that these revisions will better enable us to convey the benefits of the proposed SRA to the ICLR community.

Thank you very much,

Authors.

---

### Author Response · Authors · 2025-12-02
**Summary of the Score Changes and Discussions in Rebuttal**

Dear Area Chair,

We understand that due to the recent system incident, all scores have been reverted to their pre-rebuttal state.  Before the system freezed, our submission had effectively **moved from a mixed state (6, 4, 4, 6) to a positive state  (6, 6, 6, 6)**, with two reviewers explicitly confirming score increases and one  reviewer maintain positive score.  To assist in your decision-making:

First, we would like to highlight the position of our work: In this study, we propose SRA, improving diffusion transformers training by aligning their own representations in different layers with different noise to achieve representation alignment without external component. Removing the dependence on an external component makes our approach more general, e.g., video generation.

Then, we summarize the  score changes and the discussion during rebuttal.

- Reviewer YcnN mainly commented on some implementation，such as the hyperparameter searching and GPU usage. We gave detailed hyperparameter  setting principals and reported additional results to address these questions in the rebuttal.  In the discussion period, the reviewer explicitly stated ***"My concerns have been adequately addressed. (Hyperparameter searching, Gpu usage, Compatibility with REPA) Therefore, I will maintain my positive score."***
-  Reviewer  NzJf questioned mainly about some additional discussions and experiments，such as EMA teacher and text-to-image experiment.  We provied detailed explanations and addtional supportive experiments in rebuttal.  In the discussion period, the reviewer explicitly stated ***"Thank you very much to the authors for the rebuttal with additional experiments and discussions. Most of my concerns are resolved and raise my score to 6."***
-  Reviewer  1H2k questioned mainly about clarification and generalization of our method.  We corrected expression in our submission and provided  supportive experiments on text-to-image generation to solve the problem.  In the discussion period, the reviewer explicitly stated ***"Thank you for the rebuttal. The replies have addressed my concerns. Thus, I will raise the rating to 6."***
-  Reviewer  MZU4 questioned mainly about principled foundation and limitation discussion.  We
gave detailed hyperparameter  setting principals and discussed the limitations and the future work in rebuttal.  ***Although we did not get feedback from the reviewer before the system freezed, we believe that we provide convincing feedback to address the questions.***

We sincerely appreciate the additional time and effort you have dedicated to reviewing and assessing our submission during this special period. We also hope these explicitly stated outcomes would be considerated when making the final recommendation.

Best,

Authors

---

### Meta-Review · Area_Chair_jjKt · 2026-01-07

**Summary:**

The big concerns were mostly about the generality of the setup. Reviewers liked the idea and results, but flagged sensitivity to layer/timestep choices and other hyperparams, whether the EMA teacher adds too much compute, and whether key design bits (projection head, EMA decay, lambda behavior) feel a bit ad hoc. There were also concerns about some claims not being backed by evidence (teacher improves over time, keeps improving with longer training) and about motivating "no strong external encoder" scenarios.

Consdiering everything, the AC recommends to accept the paper.

**Reviewer Concerns:**

YcnN’s concerns (tuning guidance, GPU/memory numbers, REPA compatibility) were explicitly resolved;
NzJf’s EMA sensitivity, teacher-improves evidence (linear probing over iterations), longer training behavior, and text-to-image transfer were enough for them to raise to 6.
1H2k’s confusion about "coarse-to-fine" and why late layers can hurt was clarified and they raised to 6.
What’s still a bit open is the core motivation about settings without strong external encoders, plus criticism of “this is still mostly heuristic".

**Reviewer Scores:**

YcnN: likely stays at 6 (they said concerns were handled).
NzJf: 4 to 6 since they explicitly raised.
1H2k: 4 to 6, explicitly raised.
MZU4: most likely stays at 6. the rebuttal directly addresses their "arbitrary choices / too many knobs / missing limitations" points with extra ablations and a limitations section.

---

### Decision · Program_Chairs · 2026-01-26

Accept (Poster)